# Rethinking cross entropy for continual fine-tuning: policy gradient with entropy annealing

## Abstract

While large pretrained vision models have achieved widespread success, their post-training adaptation in continual learning remains vulnerable to catastrophic forgetting. We challenge the conventional use of cross-entropy (CE) loss, a surrogate for 0-1 loss, by reformulating classification through reinforcement learning. Our approach frames classification as a one-step Markov Decision Process (MDP), where input samples serve as states, class labels as actions, and a fully observable reward model is derived from ground-truth labels. From this formulation, we derive Expected Policy Gradient (EPG), a gradient-based method that directly minimizes the 0-1 loss (i.e., misclassification error). Theoretical and empirical analyses reveal a critical distinction between EPG and CE: while CE encourages exploration via high-entropy outputs, EPG adopts an exploitation-centric approach, prioritizing high-confidence samples through implicit sample weighting. Building on this insight, we propose an adaptive entropy annealing strategy (aEPG) that transitions from exploratory to exploitative learning during continual adaptation of a pre-trained model. Our method outperforms CE-based optimization across diverse benchmarks (Split-ImageNet-R, Split-Food101, Split-CUB100, CLRS) and parameter-efficient modules (LoRA, Adapter, Prefix). More broadly, we evaluate various entropy regularization methods and demonstrate that lower entropy of the output prediction distribution enhances adaptation in pretrained vision models. These findings suggest that excessive exploration may disrupt pretrained knowledge and establish exploitative learning as a crucial principle for adapting foundation vision models to evolving classification tasks.

## 1 Introduction

Modern vision models are prone to catastrophic forgetting when trained on non-stationary data. Traditional continual learning (CL) methods address this through memory replay mechanisms [5]. However, with the rise of large-scale pretrained vision transformers, research has shifted toward parameter-efficient fine-tuning (PEFT), which freezes most pretrained parameters and updates only a small subset of adaptable parameters. PEFT achieves state-of-the-art CL performance without relying on data replay [29, 28, 25]. Recent research has integrated various parameter injection techniques into continual learning, such as prompts, LoRA, and adapters, and proposed CL strategies like EMA ensembles and subspace initialization [9, 16] to further reduce forgetting. However, these approaches all rely on cross-entropy loss for optimization in classification tasks.

In this work, we rethink the conventional use of cross-entropy loss by reformulating continual learning through a reinforcement learning (RL) lens. The ultimate goal of classification is to minimize the misclassification error (0-1 loss), but this objective is non-differentiable and discontinuous, making it incompatible with gradient-based optimization. As a result, cross-entropy loss has become the de

Submitted to 39th Conference on Neural Information Processing Systems (NeurIPS 2025). Do not distribute.

facto surrogate for training models, even though they are ultimately evaluated on 0-1 loss. Instead, we propose directly optimizing the 0-1 loss using reinforcement learning. To achieve this, we reformulate classification as a Markov Decision Process (MDP): input samples serve as states, predicted class labels as actions, and the reward function is defined as 1 for the correct label and 0 otherwise. This formulation yields an RL objective that maximizes classification accuracy over the model's policy distribution, provably equivalent to minimizing 0-1 loss. To solve this, we introduce Expected Policy Gradient (EPG), a low-variance variant of the REINFORCE [30] policy gradient method.

Our gradient analysis reveals an interesting relationship between EPG and cross-entropy optimization: 1) Gradient alignment: EPG and CE share the same gradient direction for individual samples; 2) Sample weighting: EPG implicitly incorporates a sample-weighting mechanism that prioritizes easier samples, i.e., those where the model already exhibits high prediction confidence. This distinction also manifests in their entropy dynamics: RL optimization consistently produces output distributions with *lower entropy* than those trained with cross-entropy. Building on this insight, we propose an adaptive entropy annealing strategy (adaptive EPG): starting with CE to encourage exploration and gradually shifting toward exploitative learning (EPG). Empirically, this approach demonstrates superior performance across four continual learning benchmarks and multiple parameter-efficient training architectures (LoRA, Adapter, and Prefix-tuning).

More broadly, we investigate entropy regularization strategies for continual fine-tuning: While prior research advocates high-entropy techniques (e.g., label smoothing, focal loss, and confidence penalty) to improve classification performance, we demonstrate that these approaches actually harm performance in class-incremental learning with pretrained vision transformers. In contrast, techniques with lower entropy consistently enhance continual fine-tuning results (Table 2). This result implies that aggressive exploration can destabilize a pretrained model's learned knowledge, positioning exploitative learning as a critical strategy for continual learning with foundation models.

Our contributions are summarized as follows:

- We introduce Expected Policy Gradient (EPG), a gradient-based reinforcement learning method that directly optimizes the 0-1 loss instead of a surrogate objective, e.g. CE.

- We conduct theoretical and empirical studies revealing EPG's exploitative nature (vs. cross-entropy's exploration bias) through analysis on the gradient, entropy, and objective function (see Fig 1 and Proposition 2).

- We propose an adaptive entropy annealing strategy (aEPG) that combines the strengths of EPG and cross-entropy, achieving state-of-the-art performance in continual fine-tuning, as shown in Table 1 and 2.

- We provide evidence showing that lower entropy, contrary to traditional classification literature, improves continual adaptation of pretrained vision models (see Fig 2).

## 2 Related work

**Continual learning and parameter-efficient finetuning (PEFT)** Parameter-efficient fine-tuning techniques have recently been used in continual learning, achieving state-of-the-art performance without the need for data replay. Early work in continual fine-tuning focused on learnable prompt parameters [28, 29, 25], maintained in memory. These approaches optimize prompts to guide model predictions while explicitly managing task-invariant and task-specific knowledge. Recent advances include unified frameworks combining adapters, LoRA, and prefix tuning [9], ensemble models with online/offline PEFT experts, and specialized LoRA initialization techniques to reduce task interference [16].

**RL for fine-tuning LLMs**. RL has become pivotal for aligning large pretrained models with human preferences [2]. In the RL from human feedback (RLHF) framework, human feedback serves as the reward signal of MDP, and the model is optimized as a policy via policy gradient methods like PPO [21]. While reinforcement learning has proven highly effective for fine-tuning LLMs in generative tasks, its application to vision models and classification remains underexplored.

**RL for continual learning** Reinforcement learning has also been applied to improve continual learning performance. For instance, [31] employs RL to dynamically select optimal neural architectures for incoming tasks, while [32] introduces a multi-armed bandit framework with bootstrapped policy

gradient to adapt augmentation strength and training iterations in online continual learning. Similarly, [18] proposes a bandit-based method for online hyperparameter optimization in offline continual learning. However, these approaches focus on tuning hyperparameters rather than directly optimizing classification model parameters.

**Entropy regularization**. Entropy regularization is widely used in machine learning to influence the behavior of learned policies or predictions. 1) *Increasing entroy*: In reinforcement learning, entropy regularization encourages exploration by preventing premature convergence to suboptimal deterministic policies. Recent work by [3] applies this idea to continual RL, evaluating it on tasks such as Gridworld, CARL, and MetaWorld. Similarly, in supervised learning, entropy regularization mitigates overconfident predictions by promoting high-entropy output distributions. For instance, [22] introduces the confidence penalty, which subtracts a weighted entropy term from the loss function to produce more balanced predictions. Later, [19] unifies the understanding of label smoothing and confidence penalties, comparing their effectiveness in language generalization tasks. Additionally, [20] shows that focal loss implicitly increases entropy, improving model calibration. 2) *Decreasing entropy*. Conversely, entropy reduction is useful when training with unlabeled data and has been applied in the areas of semi-supervised learning, self-supervised learning and test-time adaptation[10]. Our work investigates entropy regularization in continual learning, particularly when pretraining from large vision models.

**Direct minimization of 0-1 loss**. Prior works have explored optimizing 0-1 directly via approximations and alternative formulations. [11] proposes a smooth approximation using the posterior mean of a generalized Beta-Bernoulli distribution. [14] employs stochastic prediction with probabilistic embeddings, modeling predictions as a multivariate normal distribution and solving optimization via orthant integration of its probability density function. Unlike these approaches, our work studies the 0-1 loss from a reinforcement learning perspective.

**Classification with bandit feedback** Our work differs from classification with bandit feedback, a problem setting introduced by [13]. In the bandit feedback setting, the learner does not observe the true label for a given input but only receives binary feedback indicating whether its predicted label was correct. And this is typically studied in an online setting and the main objective is to minimize the regret and most works investigate the properties of the hypothesis class that allow for sublinear regret [7, 8, 4]. In this paper, we explore a one-step MDP (similar to contextual bandit) framework to model a standard supervised learning problem, rather than operating under the bandit feedback setting.

# 3 Methodology

## 3.1 Problem setting: Classification as a One-Step MDP

We formulate classification and continual fine-tuning as a one-step MDP: the input samples $x \sim d(x)$ form the state space with state distribution $d(x)$; and classification labels constitute the action space $\mathcal{A}$, with a reward function $r \sim \mathcal{R}_{x,a}$ indicating whether an action (predicted label) matches the ground-truth label for $x$, or not. Episodes terminate after one step. The policy $\pi_\theta(a|x)$ is parameterized by a deep neural network. The objective is to maximize expected reward over the policy:

$$J_\pi(\theta) = \mathbb{E}_{x \sim d(x), a \sim \pi_\theta(a|x)}[r] = \sum_{x \in \mathcal{X}} d(x) \sum_{a \in \mathcal{A}} \pi_\theta(a|x)\mathcal{R}_{x,a} \tag{1}$$

More specifically, we define a deterministic reward function based on ground-truth labels:

$$\mathcal{R}_{x,a} = \begin{cases} 1, & \text{if } a = y \\ 0, & \text{otherwise} \end{cases} \tag{2}$$

This reward scheme assigns a value of one for correct classifications and zero otherwise, directly aligning the reinforcement learning objective with the goal of maximizing classification accuracy.

We focus on the supervised classification setting, where ground truth labels are available during training. This means that the reward model is fully observable to the learning agent. This differs from the problem setting of classification under bandit feedback [13].

**Connection between RL Objective and 0-1 Loss**. We establish the relationship between the reinforcement learning objective and the 0-1 classification loss. For a classifier $h_\theta$ with true labels $y$ and predictions $h_\theta(x)$, the *0-1 loss* is defined as:

$$\mathcal{L}_{01}(y, h_\theta(x)) = \begin{cases} 0, & \text{if } h_\theta(x) = y \text{ (correct prediction)} \\ 1, & \text{if } h_\theta(x) \neq y \text{ (incorrect prediction)} \end{cases} \tag{3}$$

Building upon the RL objective in Eq. 1 and the reward function in Eq. 2, we derive the following connection:

**Proposition 1.** *Minimizing the 0-1 loss of classifier $h_\theta$ is equivalent to maximizing the RL objective:*

$$\min_\theta \mathcal{L}_{01}(h_\theta) = \max_\theta J_h(\theta) \tag{4}$$

*This demonstrates that 0-1 loss minimization can be viewed as an RL problem.*

*Proof.* By interpreting $h_\theta(x)$ as the policy $\pi_\theta(a|y)$ in Eq 1 and applying a constant baseline of value 1 to the reward function $\mathcal{R}_{x,a}$ (Eq. 2), we obtain:

$$J_h(\theta) = \mathbb{E}_{x \sim d(x), a \sim h_\theta}[r] = 1 - \sum_{x \in \mathcal{X}} d(x) \sum_{a \in \mathcal{A}} h_\theta(a|x)(-\mathcal{R}_{x,a} + 1) \quad = 1 - \mathcal{L}_{01}(h_\theta) \tag{5}$$

The constant offset does not affect the optimization objective, thus establishing the equivalence. $\square$

The 0-1 loss presents fundamental challenges for gradient-based optimization due to its discontinuous and non-differentiable nature. We address this limitation through a novel reinforcement learning perspective that reformulates classification as policy optimization. While conventional classification approaches typically implement a deterministic mapping $h_\theta : \mathcal{X} \to \mathcal{Y}$ (which could alternatively be viewed as a deterministic policy in the proposed framework and optimized via deterministic policy gradient methods [24]), this paper instead explores a stochastic policy with softmax parameterization: $\pi_\theta(a|x) = e^{f_\theta(a|x)} / \sum_k e^{f_\theta(k|x)}$, where $f_\theta : \mathcal{X} \to \mathbb{R}^K$ denotes the model's logit outputs. This parameterization not only maintains the familiar structure of softmax-based classification but also establishes a principled connection between policy gradient optimization and cross-entropy minimization. Through this formulation, we can directly investigate how policy gradient methods relate to traditional classification objectives (CE) while handling the non-differentiable 0-1 loss, as shown in the next section.

## 3.2 Expected Policy Gradient

We solve the RL problem described above using policy gradient methods. While traditional approaches such as REINFORCE [30] and PPO [23] rely on stochastic action sampling, we derive a more efficient gradient estimator by exploiting the inherent structure of the classification MDP.

Based on the policy gradient theorem, the gradient of Eq 1 can be computed using the likelihood ratio gradient estimator [26]. For one-step MDPs with immediate rewards, we have:

$$\nabla_\theta J(\theta) = \mathbb{E}_{x \sim d(x), a \sim \pi_\theta(a|x)} \left[ \mathcal{R}_{x,a} \nabla_\theta \log \pi_\theta(a|x) \right] \tag{6}$$

The REINFORCE policy gradient algorithm [30] approximates this expectation through Monte Carlo sampling. Given the sampled trajectories $\{x_i, a_i, r_i\}_N$, the gradient can be estimated as:

$$\hat{g}_{\text{REINFORCE}} = \frac{1}{N} \sum_{x_i \sim d(x), a_i \sim \pi_\theta} \mathcal{R}_{x_i, a_i} \nabla_\theta \log \pi_\theta(a_i|x_i) \tag{7}$$

This type of sampling-based policy gradient method, as employed by REINFORCE and Proximal Policy Optimization (PPO) [23], is widely used in deep reinforcement learning tasks and for fine-tuning large language models with human feedback. However, we observe that the sampling-based approach does not exploit the simplicity of classification tasks. Crucially, in our classification MDP formulation, the reward function is available to the learner, since the reward $\mathcal{R}_{x,a}$ for all actions is available once the class label for a sample is given (see Eq. 2). This allows us to compute

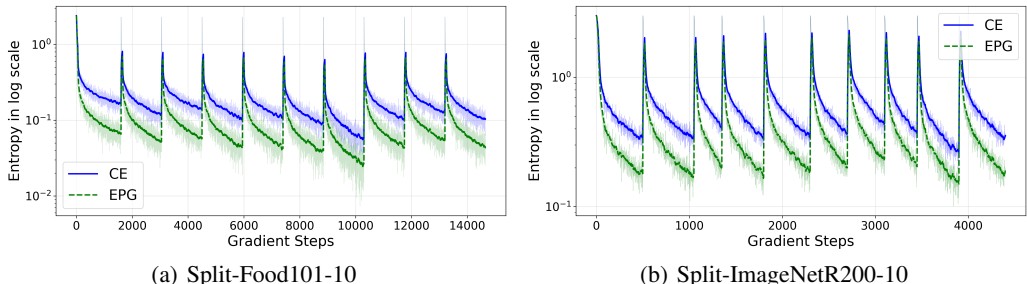

(a) Split-Food101-10                    (b) Split-ImageNetR200-10

Figure 1: The entropy of the policy, i.e. the softmax output of the model, during training. Expected policy gradient optimization (EPG) leads to higher entropy than cross-entropy (CE) optimization.

the expectation over actions exactly in the gradient estimator while only sampling from the state distribution $d(x)$:

$$\hat{g}_{\text{EPG}} = \frac{1}{N} \sum_{x_i \sim d(x)} \sum_{a \in \mathcal{A}} \pi_\theta(a|x_i) \mathcal{R}_{x_i,a} \nabla_\theta \log \pi_\theta(a|x_i) \tag{8}$$

We term this the *Expected Policy Gradient* (EPG) to distinguish it from methods that need to sample actions. As EPG uses the exact expectation, it can eliminate the noise caused by action sampling. In other words, it maintains the true gradient's expectation while reducing variance in gradient estimate, i.e., $\text{Var}[\hat{g}_{\text{EPG}}] \le \text{Var}[\hat{g}_{\text{REINFORCE}}]$, and $\mathbb{E}[\hat{g}_{\text{EPG}}] = \mathbb{E}[\hat{g}_{\text{REINFORCE}}]$.

### 3.3 Over-exploration and entropy annealing

**Connection to cross entropy**. We analyze the relationship between EPG and CE optimization. Given the target distribution $q(a|x)$ and the softmax output $\pi_\theta(a|x)$, the gradient of CE (Eq. 9) is:

$$\hat{g}_{\text{CE}} = - \sum_{x \sim d(x)} \sum_a q(a|x) \nabla_\theta \log \pi_\theta(a|x). \tag{9}$$

Note that both EPG and CE gradients involve $\nabla_\theta \log \pi_\theta(a|x)$. For one-hot labels (i.e. $q(a|x)$ follows Dirac delta distribution), the gradient for a sample $(x_i, y_i)$ simplifies to $\hat{g}_{\text{CE}}^i(x_i, y_i) = -\nabla_\theta \log \pi_\theta(y_i|x_i)$. Comparing this with Eq. 8, we derive:

$$\hat{g}_{\text{EPG}}^i(x_i, y_i) = \pi_\theta(y_i|x_i) \nabla_\theta \log \pi_\theta(y_i|x_i) = -\pi_\theta(y_i|x_i) \hat{g}_{\text{CE}}^i(x_i, y_i). \tag{10}$$

**Gradient and entropy analysis**. This reveals that EPG and CE yield gradients in the *same direction* but with *different sample weights*: EPG upweights confident predictions ($\pi_\theta(y_i|x_i) \approx 1$) while downweighting uncertain ones. To better understand how this sample weighting scheme affects gradient optimization, we analyze the entropy dynamics of both CE and EPG. Figure 1 shows the evolution of output distribution entropy during continual fine-tuning. Initially, when learning each new task, the model's predictions are nearly random, resulting in high entropy. As training progresses, this entropy gradually decreases. Perhaps surprisingly, we observe that EPG reduces entropy significantly faster than CE and achieves lower final entropy levels (Fig. 1), despite EPG having smaller gradient magnitudes than CE ($|\hat{g}_{\text{EPG}}^i(x_i, y_i)| = \pi_\theta(y_i|x_i)|\hat{g}_{\text{CE}}^i(x_i, y_i)| \le |\hat{g}_{\text{CE}}^i(x_i, y_i)|$). The entropy and gradient analysis demonstrate a key difference in their optimization behaviors: CE exhibits exploratory behavior, maintaining higher entropy in action space and promoting exploration through stochastic gradient updates that probe uncertain regions of the parameter space; EPG demonstrates exploitative tendencies, converging toward lower-entropy action distributions and more confident gradient solutions that exploit existing model knowledge.

Beyond empirical observations, we also study this phenomenon from a theoretical perspective. We demonstrate that the RL objective underlying EPG inherently minimizes entropy while simultaneously reducing the KL divergence between the target and predicted distributions (see Proposition 2).

**Proposition 2.** *For hard-label classification, the reinforcement learning objective satisfies:*

$$\max_\theta J_{p_\theta}^{RL}(\theta) \equiv \min_\theta \left[ D_{KL}(p_\theta \parallel q) + H(p_\theta) \right], \tag{11}$$

*where $p_\theta$ is the model's predictive distribution and $q$ is the target distribution. This establishes that Expected Policy Gradient optimization simultaneously minimizes the KL divergence between predictions and targets and reduces the entropy of the output distribution*

*Proof Sketch.* The equivalence follows from 1) decomposing the RL objective using the baseline subtraction technique from policy gradient methods; 2) identifying the entropy and divergence terms through algebraic manipulation The complete derivation appears in Appendix A. □

Proposition 2 reveals a fundamental connection between the 0-1 loss and KL divergence. Specifically, while the CE loss explicitly minimizes the difference between the target and predicted distributions (via minimizing $D_{\mathrm{KL}}(q||p_\theta)$), the 0-1 loss not only reduces this distributional disparity (via minimizing $D_{\mathrm{KL}}(p_\theta||q)$) but also implicitly minimizes entropy. This dual optimization mechanism provides a theoretical explanation for the empirical observation that EPG drives the model toward lower-entropy solutions compared to CE.

While prior work has shown that increased entropy can benefit classification models by promoting exploration [22, 19, 6], these advantages have primarily been observed in train-from-scratch settings. We hypothesize that this relationship may fundamentally differ for pretrained models, where excessive exploration could prove detrimental. Specifically, aggressive exploration may: 1) cause substantial deviation from the pretrained weights, compromising their inherent generalization capabilities, and 2) in continual learning settings, disrupt previously acquired task knowledge, thereby accelerating catastrophic forgetting. This necessitates a careful re-examination of the exploration-exploitation tradeoff when continually fine-tuning pretrained models.

**Adaptive entropy annealing**. To effectively balance exploration (via cross-entropy optimization) and exploitation (via expected policy gradient optimization), we propose an adaptive entropy annealing method that combines both objectives through a time-dependent weighting scheme. The combined gradient formulation is given by:

$$g_{\mathrm{aEPG}}(\theta) = \alpha_t g_{\mathrm{CE}}(\theta) + (1 - \alpha_t)(-g_{\mathrm{EPG}}(\theta)). \tag{12}$$

where $\alpha_t \in [0, 1]$ is an annealing coefficient that evolves during training. This design provides a smooth transition from initial exploration to final exploitation: beginning with pure cross-entropy optimization ($\alpha_t = 1$) to maintain high entropy during early training, we progressively shift to pure EPG ($\alpha_t = 0$) to optimize the 0-1 loss in later stages. The transition follows a sigmoid annealing schedule of $\alpha_t = \sigma\left(\tau \frac{T-2t}{T}\right)$, where $T$ represents the total number of training steps, and $\sigma(x) = (1 + e^{-x})^{-1}$ is the sigmoid function. $\tau = 6$ controls transition rate. This schedule smoothly interpolates from $\alpha_0 = \sigma(6) \approx 1$ (initialization) to $\alpha_T = \sigma(-6) \approx 0$ (convergence). Extensive experiments demonstrate that this choice of hyperparameter value is robust across diverse datasets and architectures.

## 4 Experiments

**Datasets.** We evaluate our approach on four diverse datasets spanning different image classification challenges. **ImageNet-R** [12] has renditions of 200 ImageNet classes with 24,000 training and 6,000 test samples, naturally exhibiting class imbalance. We partition it into 10 sequential tasks. **Food-101** [1] provides balanced classification across 101 food categories (750 training/250 test images per class, 101k total), split into 10 tasks. **CUB200** [27] contains 11,788 bird images across 200 species, which we organize into 10 incremental tasks (20 classes each). Finally, **CLRS** [15] offers large-scale remote sensing with 25 scene classes (600 images/class, 256×256 resolution) collected from multiple sensors, divided into 5 sequential tasks.

**Baselines.** We compare our approach (adaptive EPG, aEPG) against four state-of-the-art continual fine-tuning methods for pretrained vision transformers: Dual Prompt, LAE, InferLoRA, and standard LoRA. **DualPrompt** [28] is one of the early works that use parameter-efficient fine-tuning in continual learning by optimizing learnable prompts stored in memory. **LAE** [9] is a recent work which employs an ensemble of online and offline PEFT experts (labelled as d-lora), and **InferLoRA** [16], mitigates

Table 1: Performance comparison of continual PEFT methods on Split-ImageNet-R datasets.

| Tasks | | | 5 Tasks | | 10 Tasks | | 20 Tasks | |
|---|---|---|---|---|---|---|---|---|
| Methods | PEFT | Loss | $A_5$ | $\tilde{A}_5$ | $A_{10}$ | $\tilde{A}_{10}$ | $A_{20}$ | $\tilde{A}_{20}$ |
| DualPrompt | prefix | CE | $72.9 \pm 0.3$ | $76.0 \pm 0.3$ | $71.2 \pm 0.1$ | $75.4 \pm 0.1$ | $71.2 \pm 0.1$ | $74.8 \pm 0.1$ |
| InferLoRA | i-lora | CE | $76.8 \pm 0.4$ | $80.8 \pm 0.3$ | $74.2 \pm 0.1$ | $79.5 \pm 0.2$ | $68.6 \pm 0.5$ | $74.8 \pm 0.4$ |
| LoRA | lora | CE | $74.8 \pm 0.0$ | $79.8 \pm 0.1$ | $74.3 \pm 0.1$ | $79.2 \pm 0.3$ | $73.2 \pm 0.1$ | $78.7 \pm 0.0$ |
| LAE | d-lora | CE | $76.1 \pm 0.2$ | $80.6 \pm 0.1$ | $75.4 \pm 0.0$ | $79.9 \pm 0.3$ | $73.9 \pm 0.3$ | $79.2 \pm 0.1$ |
| LoRA + aEPG | lora | aEPG | $77.2 \pm 0.0$ | $81.9 \pm 0.0$ | $75.8 \pm 0.3$ | $80.9 \pm 0.0$ | $74.1 \pm 0.2$ | $79.7 \pm 0.2$ |
| LAE + aEPG | d-lora | aEPG | $\mathbf{78.3 \pm 0.1}$ | $\mathbf{82.3 \pm 0.0}$ | $\mathbf{76.7 \pm 0.3}$ | $\mathbf{81.4 \pm 0.2}$ | $\mathbf{75.0 \pm 0.3}$ | $\mathbf{80.0 \pm 0.1}$ |

task interference via carefully initialized LoRA subspaces (labeled as i-lora). We also evaluate standard **LoRA**, applied in continual fine-tuning with local cross-entropy loss; this baseline has been shown to outperform DualPrompt [9]. Our experiments adopt the unified framework of [9], which supports diverse PEFT methods, including Adapter, LoRA, and Prefix.

We further compare EPG and aEPG against entropy regularization techniques, including: **Focal loss** [17], **Label smoothing** [19], and **Confidence penalty** [22]. Following previous CL works [9, 16], all losses are applied locally in the continual fine-tuning experiments, computed exclusively over the current task's categories.

**Training details.** We adopt a ViT-B/16 backbone (vit_base_patch16_224 from timm library), pre-trained on ImageNet-21k and fine-tuned on ImageNet-1k. All experiments use PyTorch with the Adam optimizer (learning rate=0.0005, batch size=256,). We initialize classifier heads from $\mathcal{N}(0, 0.001)$, an aspect previously uncontrolled in the release code of previous continual fine-tuning works. Following [9], the ViT backbone remains frozen for the first 30 epochs before full fine-tuning for 20 additional epochs (50 total). All PEFT modules (LoRA, Adapters, or Prefix Tuning) are applied to the first 5 transformer blocks (results for 10 blocks show a similar pattern and are omitted), with LoRA configured to rank 4. Unless otherwise specified, we report mean performance metrics with standard deviations across 5 independent runs with different random seeds.

**Evaluation metrics.** We evaluate all models with a widely used incremental metric: the end accuracy on all the seen tasks $A_T = 1/T \sum_{i=1}^{i=T} a_{i,T}$ , where $T$ is the total number of tasks and $a_{i,j}$ denotes the accuracy of the $j$-th task once the model has learned the $t$-th task. We also report the average accuracy $\tilde{A}_T = \frac{1}{T} \sum_{t=1}^{t=T} A_t$.

## 4.1 Results

**Continual fine-tuning results.** Our first evaluation is based on ImageNet-R with 5, 10, and 20 task splits. Table 1 demonstrates aEPG outperforming the baseline methods. In addition, unlike DualPrompt or InferLoRA, which rely on a specific architectural design, our method introduces a novel loss formulation that can be easily combined with different continual learning frameworks. Table 1 demonstrates that aEPG can be combined with LAE to achieve the best results.

We further validate aEPG's effectiveness through comprehensive experiments on Split-ImageNetR200, Split-Food101, Split-CUB200, and CLRS datasets, demonstrating consistent improvements over cross-entropy optimization across diverse post-training architectures, including LoRA, Adapter, and Prefix tuning (see Table 2).

**Entropy dynamics**. Inspired by the promising results of aEPG in Table 1, we systematically investigate the effects of increasing or decreasing entropy during fine-tuning of pretrained models. To increase entropy, we employ established methods such as label smoothing, confidence penalty, and focal loss. Conversely, to decrease entropy, we leverage EPG and aEPG, which have been shown to reduce entropy in Section 3.3 and Fig. 1. Additionally, we evaluate an entropy-penalized (EP) loss adapted from CE, structured similarly to Proposition 2. This objective simultaneously minimizes the KL divergence and entropy:

$$\min \mathcal{L}_{EP} = \min \left[ \mathcal{L}_{CE} + H(p_\theta) \right] = \min \left[ D_{\text{KL}}(q || p_\theta) + H(p_\theta) \right]$$

The objective functions are summarized in Table 4 in the Appendix.

Fig. 2 illustrates the entropy dynamics and continual learning performance. Generally, entropy-increasing methods (focal loss, label smoothing, confidence penalty) degrade performance, whereas

Table 2: Algorithm performance comparison across four datasets and different PEFT modules.

| PEFT | Algo | $H(p_\theta)$ | Split-ImageNetR200 | | Split-Food101 | | Split-Cub200 | | CLRS25 | |
|---|---|---|---|---|---|---|---|---|---|---|
| | | | $A_{10}$ | $\tilde{A}_{10}$ | $A_{10}$ | $\tilde{A}_{10}$ | $A_{10}$ | $\tilde{A}_{10}$ | $A_5$ | $\tilde{A}_5$ |
| LoRA | CE | - | $74.1 \pm 0.4$ | $79.3 \pm 0.3$ | $83.2 \pm 0.2$ | $88.8 \pm 0.2$ | $83.3 \pm 0.2$ | $86.2 \pm 0.1$ | $74.2 \pm 0.8$ | $83.9 \pm 0.2$ |
| | Focal | ↑ | $72.4 \pm 0.3$ | $77.9 \pm 0.3$ | $82.7 \pm 0.3$ | $88.4 \pm 0.2$ | $82.9 \pm 0.3$ | $86.1 \pm 0.2$ | $73.0 \pm 0.9$ | $82.5 \pm 0.4$ |
| | LS | ↑ | $70.6 \pm 0.2$ | $76.7 \pm 0.2$ | $77.5 \pm 0.4$ | $85.2 \pm 0.2$ | $82.9 \pm 0.2$ | $86.6 \pm 0.2$ | $74.8 \pm 1.1$ | $85.1 \pm 0.5$ |
| | CP | ↑ | $72.4 \pm 0.8$ | $77.9 \pm 0.7$ | $83.0 \pm 0.2$ | $88.9 \pm 0.2$ | $83.3 \pm 0.3$ | $86.5 \pm 0.2$ | $74.0 \pm 1.0$ | $83.9 \pm 0.3$ |
| | EPG | ↓ | $75.1 \pm 0.4$ | $80.0 \pm 0.2$ | $83.5 \pm 0.3$ | $88.9 \pm 0.2$ | $84.2 \pm 0.1$ | $85.9 \pm 0.1$ | $74.6 \pm 0.8$ | $84.3 \pm 0.7$ |
| | aEPG | ↓ | $\mathbf{75.5 \pm 0.1}$ | $\mathbf{80.9 \pm 0.1}$ | $\mathbf{84.4 \pm 0.1}$ | $\mathbf{89.5 \pm 0.1}$ | $84.7 \pm 0.3$ | $86.7 \pm 0.1$ | $\mathbf{76.3 \pm 0.4}$ | $\mathbf{85.5 \pm 0.3}$ |
| | EP | ↓ | $75.1 \pm 0.2$ | $80.4 \pm 0.3$ | $84.0 \pm 0.2$ | $89.2 \pm 0.2$ | $\mathbf{85.0 \pm 0.2}$ | $\mathbf{87.2 \pm 0.1}$ | $74.8 \pm 0.8$ | $84.6 \pm 0.5$ |
| Adapter | CE | - | $73.7 \pm 0.2$ | $79.4 \pm 0.1$ | $82.9 \pm 0.1$ | $88.5 \pm 0.1$ | $83.7 \pm 0.3$ | $86.3 \pm 0.2$ | $75.6 \pm 1.2$ | $83.7 \pm 0.9$ |
| | Focal | ↑ | $72.1 \pm 0.2$ | $77.9 \pm 0.2$ | $82.4 \pm 0.2$ | $88.1 \pm 0.1$ | $82.7 \pm 0.3$ | $86.2 \pm 0.3$ | $73.2 \pm 0.9$ | $82.7 \pm 1.0$ |
| | LS | ↑ | $70.7 \pm 0.7$ | $77.2 \pm 0.5$ | $77.3 \pm 0.3$ | $85.1 \pm 0.2$ | $83.2 \pm 0.3$ | $86.2 \pm 0.1$ | $75.5 \pm 1.0$ | $84.9 \pm 0.8$ |
| | CP | ↑ | $71.8 \pm 0.4$ | $77.9 \pm 0.1$ | $83.5 \pm 0.2$ | $89.1 \pm 0.1$ | $84.1 \pm 0.2$ | $87.0 \pm 0.2$ | $74.5 \pm 0.7$ | $83.6 \pm 0.9$ |
| | EPG | ↓ | $75.1 \pm 0.3$ | $80.3 \pm 0.2$ | $83.5 \pm 0.3$ | $88.8 \pm 0.1$ | $84.7 \pm 0.3$ | $86.4 \pm 0.2$ | $77.5 \pm 1.1$ | $84.9 \pm 1.2$ |
| | aEPG | ↓ | $\mathbf{75.4 \pm 0.2}$ | $\mathbf{81.3 \pm 0.1}$ | $\mathbf{84.4 \pm 0.1}$ | $\mathbf{89.4 \pm 0.1}$ | $85.0 \pm 0.2$ | $86.9 \pm 0.2$ | $\mathbf{77.9 \pm 0.6}$ | $\mathbf{85.5 \pm 1.3}$ |
| | EP | ↓ | $74.8 \pm 0.2$ | $80.4 \pm 0.1$ | $83.8 \pm 0.1$ | $89.1 \pm 0.1$ | $\mathbf{85.4 \pm 0.4}$ | $\mathbf{87.3 \pm 0.2}$ | $76.6 \pm 0.8$ | $85.2 \pm 0.5$ |
| Prefix | CE | - | $73.5 \pm 0.2$ | $77.8 \pm 0.2$ | $82.9 \pm 0.2$ | $88.8 \pm 0.2$ | $81.7 \pm 0.3$ | $85.1 \pm 0.2$ | $70.7 \pm 1.3$ | $80.6 \pm 0.9$ |
| | Focal | ↑ | $72.2 \pm 0.3$ | $76.7 \pm 0.3$ | $82.6 \pm 0.4$ | $88.4 \pm 0.3$ | $81.6 \pm 0.2$ | $85.5 \pm 0.2$ | $68.9 \pm 1.3$ | $79.0 \pm 1.2$ |
| | LS | ↑ | $69.9 \pm 1.2$ | $74.7 \pm 1.2$ | $77.2 \pm 0.4$ | $85.5 \pm 0.3$ | $82.9 \pm 0.5$ | $86.4 \pm 0.6$ | $\mathbf{74.4 \pm 0.7}$ | $\mathbf{83.3 \pm 0.3}$ |
| | CP | ↑ | $70.4 \pm 0.2$ | $74.8 \pm 0.2$ | $83.7 \pm 0.1$ | $89.2 \pm 0.1$ | $83.0 \pm 0.3$ | $\mathbf{86.7 \pm 0.2}$ | $71.7 \pm 1.2$ | $81.2 \pm 0.9$ |
| | EPG | ↓ | $74.7 \pm 0.1$ | $78.7 \pm 0.3$ | $83.2 \pm 0.4$ | $88.8 \pm 0.1$ | $82.3 \pm 0.2$ | $84.8 \pm 0.2$ | $74.1 \pm 1.0$ | $82.6 \pm 0.5$ |
| | aEPG | ↓ | $\mathbf{75.2 \pm 0.1}$ | $\mathbf{79.2 \pm 0.1}$ | $\mathbf{84.2 \pm 0.2}$ | $\mathbf{89.5 \pm 0.1}$ | $82.4 \pm 0.2$ | $85.3 \pm 0.2$ | $73.7 \pm 0.8$ | $82.6 \pm 0.5$ |
| | EP | ↓ | $74.6 \pm 0.1$ | $78.7 \pm 0.1$ | $83.9 \pm 0.2$ | $89.3 \pm 0.1$ | $\mathbf{83.0 \pm 0.2}$ | $85.8 \pm 0.2$ | $73.1 \pm 0.7$ | $82.0 \pm 0.4$ |

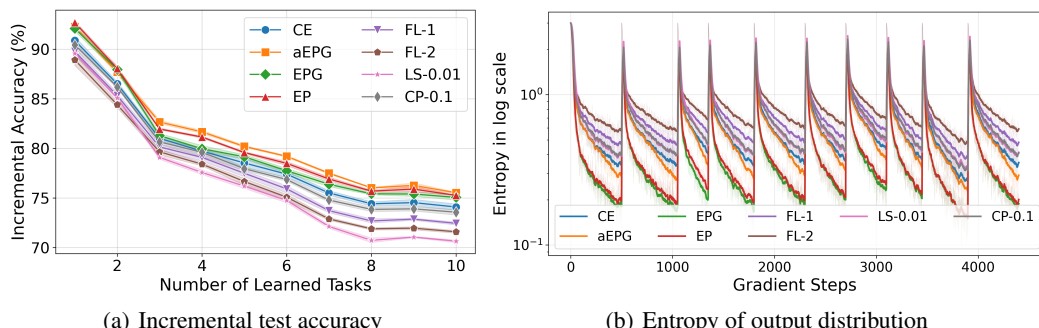

(a) Incremental test accuracy

(b) Entropy of output distribution

Figure 2: Entropy dynamics in continual fine-tuning of VisionTransformers on Split-ImagenetR200. Compared to the cross-entropy loss, Expected Policy Gradient (EPG), adaptive EPG (aEPG), and Entropy Penalty (EP) lead to lower entropy and improved accuracy. In contrast, focal loss, label smoothing, and confidence penalty (CP) lead to higher entropy and worse performance. Results for Split-Food101 datasets can be found in Appendix D.2

entropy-decreasing methods (EPG, aEPG, EP) improve it. Detailed quantitative results with optimal hyperparameters are provided in Table 2. Notably, we observe that aEPG achieves the best performance on ImageNet-R, Food101, and CLRS, and EP performs best on CUB200. All entropy-reducing methods (EPG, aEPG, EP) outperform the cross-entropy baseline.

## 4.2 Ablation studies

**Effect of $\alpha$ on objective combination.** We analyze the impact of the weighting coefficient $\alpha$ when combining cross-entropy $\mathcal{L}_{ce}(\theta)$ and the reinforcement learning objective $-J(\theta)$. Fig. 3 compares performance across $\alpha \in [0.0, 0.1, 0.2, 0.5, 0.7, 1.0]$. We observe that: 1) lower $\alpha$ values (emphasizing the RL objective) generally produce superior results, and 2) our adaptive $\alpha$ scheduling strategy consistently outperforms fixed $\alpha$ configurations. These findings suggest that dynamic adjustment of the loss weighting with entropy annealing is crucial for optimal performance.

**Entropy annealing mechanism** We also explore alternative annealing schedules such as linear decay ($\alpha_t = \frac{T-t}{T}$) and cosine decay ($\alpha_t = \frac{1}{2} + \frac{1}{2} \cos \pi \frac{t}{T}$). Our analysis reveals that entropy annealing performance remains stable across different schedule choices. Alternative approaches including linear decay and cosine decay yield comparable results as shown in Fig 3.

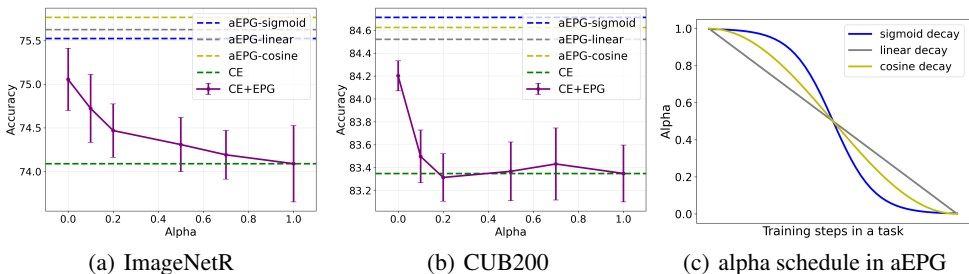

(a) ImageNetR  (b) CUB200  (c) alpha schedule in aEPG

Figure 3: The effect of alpha when combining CE and EPG with $\alpha\mathcal{L}_{CE} + (1-\alpha)\mathcal{L}_{EPG}$

Table 3: The algorithm performance for training ResNet-50 from scratch.

| Method | $\alpha = 0$ | $\alpha = 0.2$ | $\alpha = 0.5$ | $\alpha = 1$ |
|---|---|---|---|---|
| CIFAR100 | $7.32 \pm 0.70$ | $\mathbf{80.80 \pm 0.13}$ | $79.95 \pm 0.20$ | $77.95 \pm 0.78$ |
| CIFAR10 | $92.22 \pm 0.51$ | $\mathbf{95.81 \pm 0.05}$ | $95.62 \pm 0.08$ | $95.31 \pm 0.18$ |

**Train from scratch** While this work primarily focuses on continual fine-tuning, our findings that RL and EPG methods can optimize the 0-1 loss are broadly applicable to standard supervised learning. To investigate this, we trained ResNet-50 from scratch on CIFAR-10 and CIFAR-100 for 350 epochs using the optimal training and learning rate annealing schedule for cross-entropy loss as reported in the literature [22] (see Appendix C). Consistent with our finding in the continual fine-tuning experiments, EPG optimization demonstrated faster entropy convergence than CE optimization, as shown in Fig. 4c in the appendix. However, when training from a randomly initialized model, we observe that EPG's entropy decreases excessively, ultimately hindering the learning process, a phenomenon not observed when initializing from pretrained models. Interestingly, combining EPG and CE with an alpha value of 0.2 or 0.5 yields superior performance compared to CE alone, achieving performance gains of approximately 2% on CIFAR-100 and 0.5% on CIFAR-10. These results suggest that incorporating 0-1 loss into CE optimization not only benefits continual fine-tuning but also enhances standard training from scratch. A plausible explanation is that entropy reduction aligned with 0-1 loss facilitates faster convergence.

## 5  Discussion and conclusion

In this work, we re-examined the conventional use of cross-entropy loss in continual learning and proposed a novel reinforcement learning framework that directly optimizes the 0-1 misclassification error, i.e. the true objective of classification tasks. By reformulating classification as a Markov Decision Process and introducing Expected Policy Gradient (EPG), we demonstrated that RL-based optimization aligns with the ultimate goal of minimizing classification errors while exhibiting distinct gradient and entropy dynamics compared to CE. Our theoretical and empirical analyses revealed that EPG implicitly prioritizes high-confidence predictions, leading to lower-entropy output distributions and improved stability in continual fine-tuning scenarios. To bridge the gap between exploration (encouraged by CE) and exploitation (favored by EPG), we introduced an adaptive entropy annealing strategy (aEPG) that transitions smoothly from CE to EPG, achieving state-of-the-art performance across multiple CL benchmarks and parameter-efficient fine-tuning (PEFT) architectures. Furthermore, we challenged the conventional wisdom that high-entropy regularization benefits classification, showing instead that lower entropy consistently enhances class-incremental learning with pretrained vision transformers.

**Limitations**. While our method demonstrates strong performance in continual fine-tuning with vision transformers, it has several limitations. First, our theoretical and empirical analyses assume a standard supervised setting with clean, hard labels, leaving EPG's robustness to noisy or ambiguous samples an open question for future work. Second, our experiments focus exclusively on class-incremental learning with pretrained vision transformers, and further validation is needed to assess generalizability to other architectures (e.g., CNNs) or modalities (e.g., language models).

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

## A Proof of Proposition 2

$$D_{\mathrm{KL}}(p_\theta \parallel q) = \sum_{k=1}^{K} p_\theta(y_k|x) \log \frac{p_\theta(y_k|x)}{q(y_k|x)}$$

$$= \underbrace{\sum_k p_\theta(y_k|x) \log p_\theta(y_k|x)}_{-H(p_\theta)} - \sum_k p_\theta(y_k|x) \log q(y_k|x) \quad (13)$$

$$= -H(p_\theta) - \mathbb{E}_{p_\theta}[\mathcal{R}'(y_k, x)]$$

where $\mathcal{R}'(y_k, x) \doteq \log q(y_k|x)$. For Dirac delta distributions $q(y_k|x) = \delta_{y_k, y^*}$:

$$\mathcal{R}'(y_k, x) = \begin{cases} \log(1) & \text{if } y_k = y^* \\ \log(0) & \text{otherwise} \end{cases} \quad (14)$$

**Reward Baseline Adjustment.** Using the policy gradient invariance to constant baselines, we set $A \doteq -\log(0)$ and define:

$$\mathbb{E}_{p_\theta}[\mathcal{R}'] = -A + \mathbb{E}_{p_\theta}[\mathcal{R}' + A]$$
$$= -A + A \cdot \mathbb{E}_{p_\theta}[\mathcal{R}] \quad (15)$$

where $\mathcal{R}(y_k, x)$ is the 0-1 reward:

$$\mathcal{R}(y_k, x) = \begin{cases} 1 & \text{if } y_k = y^* \\ 0 & \text{otherwise} \end{cases} \quad (16)$$

**Final Equivalence.** Substituting (15) into (13) yields:

$$D_{\mathrm{KL}}(p_\theta \parallel q) = -H(p_\theta) - A \cdot \mathbb{E}_{p_\theta}[\mathcal{R}] + A \quad (17)$$

Since $A > 0$ is constant, maximizing the expected reward is equivalent to:

$$\max_\theta \mathbb{E}_{p_\theta}[\mathcal{R}] \equiv \min_\theta \left( D_{\mathrm{KL}}(p_\theta \parallel q) + H(p_\theta) \right) \quad (18)$$

**Generalization to Label Smoothing.** The same equivalence holds when $q$ follows a uniform label smoothing distribution, with the proof following analogous steps by substituting $q(y|x) = \epsilon/K + (1-\epsilon)\delta_{y,y^*}$, where $K$ is the number of classes and $\epsilon$ controls the smoothing intensity.

## B Loss function details

Table 4 compares the objective functions of different entropy regularization methods. Focal loss, label smoothing, and confidence penalty increase entropy, whereas EPG, aEPG, and entropy penalty reduce it.

Table 4: Loss functions and their effects on entropy

| Training Method | Loss Function | Entropy |
|---|---|---|
| Cross Entropy | $L_{CE} = -\sum q \log p_\theta$ | Baseline |
| Confidence Penalty | $L_{CP} = L_{CE} - \beta H(p_\theta)$ | |
| Label Smoothing | $L_{LS} = (1-\gamma)L_{CE} + \gamma D_{\mathrm{KL}}(u\|p_\theta)$ | Increase entropy ↑ |
| Focal loss | $L_{FL} = (1-p_\theta)^\gamma L_{CE}$ | |
| Expected Policy Gradient | $L_{EPG} = -\mathbb{E}_{p_\theta}[q]$ | |
| Entropy penalty | $L_{EP} = L_{CE} + H(p_\theta)$ | Decrease entropy ↓ |
| aEPG | $L_{aEPG} = \alpha_t L_{CE} + (1-\alpha_t)L_{EPG}$ | |

Table 5: Hyperparameter setting in the continual fine-tuning experiments.

| Hyperparameters | Settings |
|---|---|
| Pretrained model | vit_base_patch16_224 |
| Training epoch | 50 |
| Backbone freeze epoch | 30 |
| Batch size | 256 |
| Learning rate | 0.0005 |
| Optimizer | Adam ($\beta1 = 0.9, \beta2 = 0.999$, eps=1e-08) |
| Weight decay | 0 |
| Gradient clipping | None |
| Classifier initialization | Normal distribution with std of 0.001 |
| Augmentation | Random Resized Crop: scale = (0.05, 1.0), ratio = (3. / 4., 4. / 3.), Random Horizontal Flip (p=0.5) |
| Focal loss | gamma: 0.5,1,2 |
| Label smoothing | smooth parameter: 0.01, 0.05, 0.1 |
| Confidence penalty | penalty intensity: 0.1,0.2 |
| aEPG | tau = 6 |
| EP | beta = 1 |
| Dualprompt | $L_g = 5, L_e = 20$ |
| InferLoRA | $\epsilon = 1e - 8$, lamb=0.99, lame=1.0, rank=5 |
| LAE | EMA decay: 0.999 |
| LoRA | block: [0-4], rank = 4 |
| Adapter | block: [0-4], down_sample = 5 |
| Prefix | block: [0-4], length = 10 |

## C  Implementation details

**Continual fine-tuning experiments**. We evaluate all methods using consistent pretraining weights[1] and optimization settings. The detailed hyperparameter settings for all algorithms are listed in Table 5. For DualPrompt, InferLoRA, and LAE, we adopt the key algorithm-specific hyperparameters following their original papers and official implementations. Our implementation builds upon the LAE codebase [2]. For DualPrompt, we use the PyTorch implementation from [3], while the results for InferLoRA are based on the code released at [4].

Our experiments were conducted on NVIDIA RTX A6000 and NVIDIA A100 GPUs. The average runtime for a single dataset in one independent run ranges between 1–5 hours, depending on the task complexity.

**Train from sratch**. All models were trained for 350 epochs with a learning rate reduced by a factor of 10 at epochs 150 and 225. We used Stochastic Gradient Descent (SGD) with a batch size of 256 and momentum of 0.9. We report mean performance metrics with standard deviations across 3 independent runs with different random seeds.

## D  Additional experiment results

### D.1  Training from scratch

Figure 4 illustrates the test accuracy and entropy evolution during training on CIFAR100 and CIFAR10 from random initialization. We observe that smaller alpha values accelerate entropy convergence, with $\alpha = 0.2$ achieving optimal performance (2% improvement over standard cross-entropy). This demonstrates the advantage of combining 0-1 loss with cross-entropy. Notably, pure 0-1 optimization

---

[1]`https://storage.googleapis.com/vit_models/augreg/B_16-i21k-300ep-lr_0.001-aug_m`
`edium1-wd_0.1-do_0.0-sd_0.0--imagenet2012-steps_20k-lr_0.01-res_224.npz`

[2]`https://github.com/gqk/LAE`

[3]`https://github.com/JH-LEE-KR/dualprompt-pytorch`

[4]`https://github.com/liangyanshuo/InfLoRA`

Table 6: Train from scratch hyperparameter setting

| Hyperparameters | Setting |
| --- | --- |
| Batch size | 256 |
| Training epoch | 350 |
| LR milestone | 100,225 |
| Learning rate | 0.1,0.01,0.001 |
| Optimizer | SGD |
| Momentum | 0.9 |
| Weight decay | 0 |
| Gradient clipping | None |
| Augmentation | Random Crop: padding=4, Random Horizontal Flip (p=0.5) |

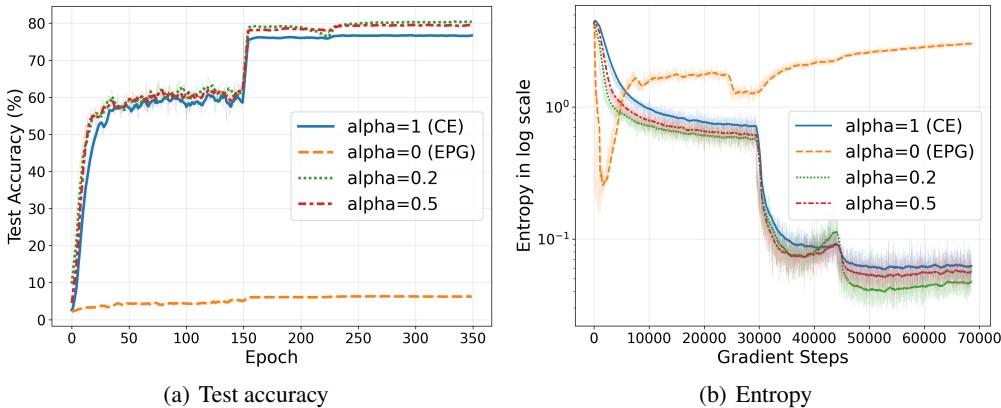

(a) Test accuracy

(b) Entropy

Figure 4: Training CIFAR100 with ResNet50 from scratch. CE-EPG with an alpha value of 0.2 outperforms the standard CE loss (Best test accuracy: 81% vs. 78%.)

($\alpha = 0.2$) fails to converge effectively for CIFAR100, unlike in pretrained models. This suggests that randomly initialized networks require stronger initial exploration.

## D.2 Continual fine-tuning results

Figure 5 illustrates the entropy dynamics on the Split-Food101 dataset, revealing trends similar to those observed on Split-ImageNetR. Compared to cross-entropy loss, Expected Policy Gradient (EPG), adaptive EPG (aEPG), and Entropy Penalty (EP) achieve lower entropy and higher accuracy. In contrast, focal loss, label smoothing, and confidence penalty (CP) result in higher entropy and degraded performance. Label smoothing is particularly detrimental: even with a small smoothing parameter (0.01), it reduces final accuracy by approximately 5%.

Figure 6 analyzes the effect of entropy regularization strength in focal loss, label smoothing, and confidence penalty. Increasing regularization typically leads to substantially higher entropy, which in turn degrades performance. For example, label smoothing with a parameter of 0.1 performs worse than with 0.01, reinforcing our observation that excessive exploration harms continual fine-tuning. The only exception is focal loss: both gamma=2 and gamma=0.5 underperform compared to gamma=1.

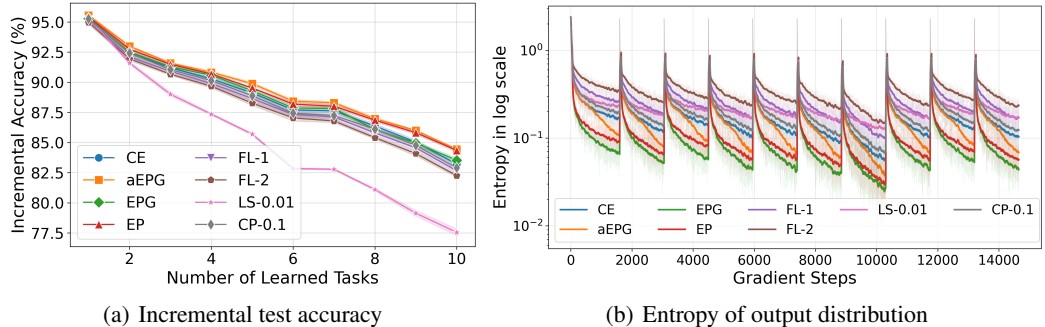

(a) Incremental test accuracy

(b) Entropy of output distribution

Figure 5: Entropy dynamics in continual fine-tuning VisionTransformers on Split-Food101. Compared to the cross-entropy loss, Expected Policy Gradient (EPG), adaptive EPG (aEPG), and Entropy Penalty (EP) lead to lower entropy and improved accuracy. In contrast, focal loss, label smoothing, and confidence penalty (CP) lead to higher entropy and worse performance.

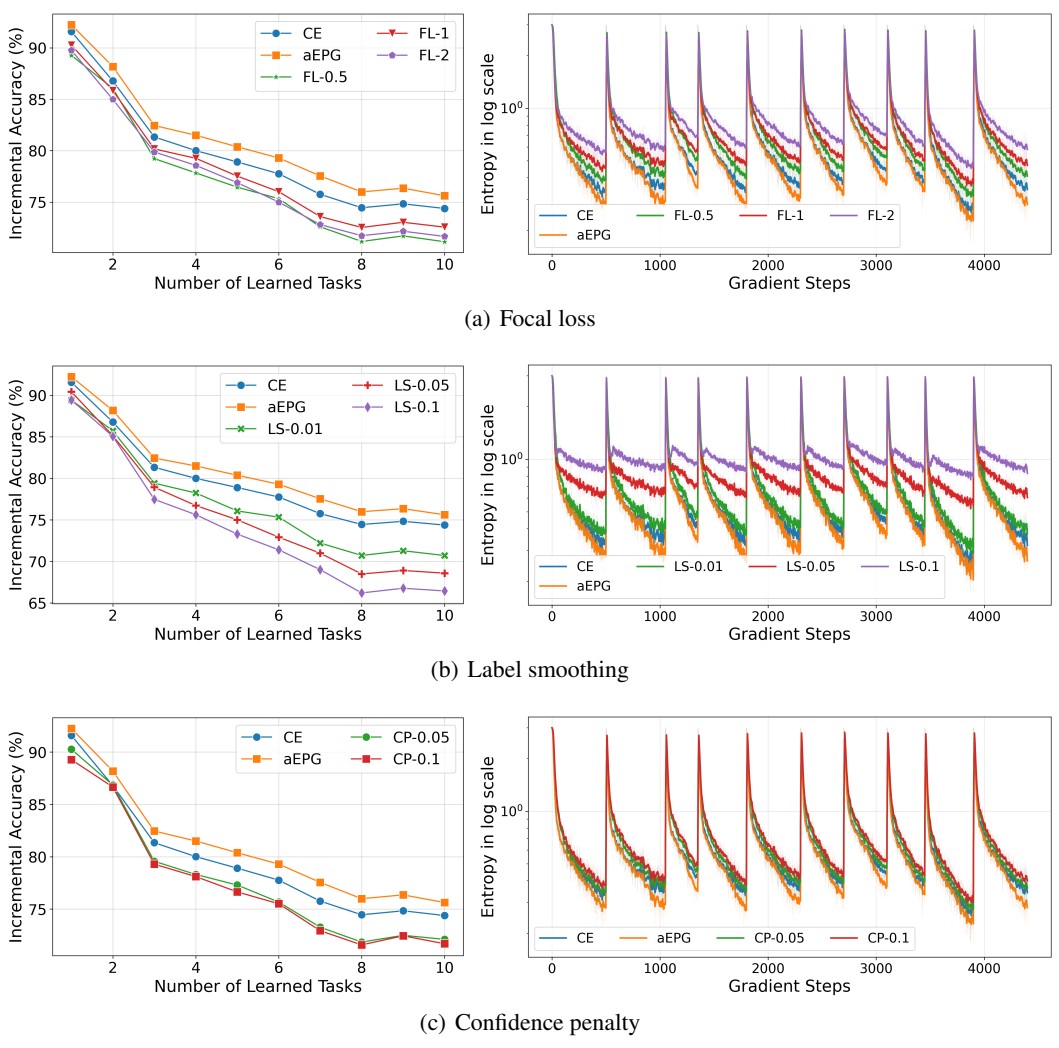

(a) Focal loss

(b) Label smoothing

(c) Confidence penalty

Figure 6: The performance of entropy regularization methods (Focal loss, label smoothing, confidence penalty) in continual fine-tuning ViT in Split-ImageNetR using different regularization strengths.

