# OpenReview forum: "Rethinking cross entropy for continual fine-tuning: policy gradient with entropy annealing"
_NeurIPS.cc/2025/Conference — Submitted to NeurIPS 2025_

### Official Review · Reviewer_L5mY · 2025-07-01

**Clarity:** 3
**Significance:** 2
**Originality:** 3
**Rating:** 3
**Confidence:** 4

**Summary:**

This paper addresses continual learning with labeled datasets. The authors propose a reinforcement learning-based formulation that treats classification as a one-step Markov Decision Process, where the input, predicted label, and correctness correspond to the state, action, and reward, respectively. They first introduce Expected Policy Gradient (EPG), which leverages the expected reward to eliminate the noise introduced by action sampling. They then present the mathematical connection between EPG and cross-entropy loss. To balance the trade-off between exploration and exploitation, they propose adaptive entropy annealing, which initially applies cross-entropy optimization and gradually transitions to EPG. Experiments under parameter-efficient fine-tuning settings demonstrate that reducing output entropy improves continual learning performance.

**Questions:**

Please address the concerns raised in Weaknesses 1, 2, and 3.

**Ethical Concerns:**

["NO or VERY MINOR ethics concerns only"]

**Final Justification:**

The authors have provided a thoughtful and sincere rebuttal, which I appreciate. My concerns are primarily related to the overall quality and I acknowledge that their proposed improvements partially address these issues. That said, I believe the initial manuscript still requires notable revisions in writing and structure. Therefore, I am finalizing my score as a borderline reject.

**Limitations:**

While this paper presents a interesting attempt to apply reinforcement learning techniques such as policy gradient to image classification, particularly in the context of continual learning, the reviewer finds that the current level of contribution falls short of what is needed to make a meaningful impact on the field. Therefore, the reviewer is inclined to recommend rejection. It is hoped that addressing the concerns raised in the weaknesses section will lead to substantial improvements in the paper.

**Paper Formatting Concerns:**

There are no paper formatting concerns.

**Quality:**

3

**Strengths And Weaknesses:**

### Strengths
1. The application of reinforcement learning to computer vision classification is new, and this direction has the potential to influence future research.
2. The comparison with widely used entropy-increasing methods, such as focal loss and label smoothing, clearly motivates the suitability of the proposed RL-based approach for continual learning.
3. The paper combines theoretical justification with empirical results effectively.

### Weaknesses
1. The paper would benefit from a more comprehensive review of prior work. For instance, the statement in Line 25 `Traditional continual learning (CL) methods address this through memory replay mechanisms [5].` may mislead readers unfamiliar with the field into thinking that all past CL methods rely on replay. In fact, there are many other CL approaches [A], including parameter regularization, knowledge distillation, and meta-learning. Several statements like this raises concerns about whether the prior research has been sufficiently surveyed and effectively communicated. A clearer categorization and broader set of citations [B] would help situate the work.
2. To increase its broader impact, the paper would benefit from additional effort in demonstrating the **scalability** of the proposed method. Despite having access to high-end hardware (A6000 and A100), the authors do not evaluate their approach on other standard computer vision tasks that also rely on cross-entropy loss, such as detection or segmentation. This omission limits the academic influence of their work. For example, MoCo [C] demonstrated how a simple loss like InfoNCE could lead to strong visual representations, validated across tasks including detection and segmentation. If the paper tackled a fundamentally novel challenge, as Mamba [D] do, then the use of relatively small benchmarks would still be justifiable. However, the reviewer remain unconvinced that the proposed approach constitutes such a breakthrough.
3. The paper would benefit from a clearer explanation of the experimental setup and its underlying motivations. The authors might imply that "Prior CL methods have adopted PEFT to improve performance, and we simply follow this trend". However, the rationale would be more convincing if framed as: "PEFT helps preserve pretrained weights, improving stability and efficiency in CL. We adopt it as a baseline. Moreover, we evaluate our method under full fine-tuning to demonstrate broader applicability." This kind of justification is missing. As a result, the reviewer find it difficult to interpret the reported gains in Table 2, approximately 0.5% improvement, as clearly meaningful. It raises the question of whether such gains could be achieved through modest hyperparameter tuning (e.g., learning rate, weight decay, batch size) rather than through the proposed method itself.

[A] Parameter-Efficient Continual Fine-Tuning: A Survey. arXiv 2025
[B] A survey and taxonomy of loss functions in machine learning. arXiv 2024
[C] Momentum Contrast for Unsupervised Visual Representation Learning. CVPR 2020
[D] Mamba: Linear-Time Sequence Modeling with Selective State Spaces. COLM 2024

---

> ### Author Rebuttal · Authors · 2025-07-31
>
> We thank the reviewer for taking the time to read and critically assess our paper.
>
> We are encouraged by the reviewer’s positive feedback, particularly the comment that our work is _“new and has the potential to influence future research.”_ We also appreciate the recognition that our paper effectively combines theoretical justification with empirical evidence. Below, we address the reviewer’s specific concerns:
>
> **1.Comparison with Related Continual Learning Works**
>
> The continual learning approaches mentioned by the reviewer—such as parameter regularization, knowledge distillation, and meta-learning—were originally developed for training-from-scratch settings, typically using full-model training with a global cross-entropy (CE) loss. These approaches are less applicable in the context of continual parameter-efficient fine-tuning (PEFT) with local cross-entropy loss.
>
> Recent empirical studies (e.g., DualPrompt [1], L2P [2]) have shown that these traditional methods often underperform when compared to modern PEFT-specific methods such as L2P and DualPrompt. For this reason, our comparisons focus on state-of-the-art continual PEFT techniques, including DualPrompt, LAE, LoRA, and InferLoRA, which are more appropriate and competitive in the continual fine-tuning paradigm we target.
>
> [1] "Dualprompt: Complementary prompting for rehearsal-free continual learning", ECCV 2022.
>
> [2] "Learning to prompt for continual learning" (L2P) CVPR 2022
>
> **2. Benchmarks, and Scope of Application**
>
> Benchmarks: We carefully reviewed our benchmark setup in relation to prior works, including DualPrompt and LAE. Our evaluation spans a broader and more challenging set of benchmarks. While LAE and DualPrompt are evaluated on two datasets (CIFAR-100 and ImageNet-R), our study includes four more demanding datasets: ImageNet-R, Food101, CUB-200, and CLRS. These additions significantly increase both the data scale (e.g., Food101) and domain diversity (e.g., CLRS, which involves remote sensing data).
>
>
> |   | Benchmarks  |
> |:---|:---|
> | DualPrompt  | ImageNet-R, CIFAR100  |
> | LAE  | ImageNet-R, CIFAR100  |
> | Ours work  | ImageNet-R, Food101, CUB200, CLRS  |
>
> **Robustness**. We further evaluated the methods under label noise conditions, demonstrating that the proposed approach remains robust and continues to outperform CE-based training under noisy supervision.
>
> **Additional Result Table 1: Evaluation in noisy label setting**
> | 20% Noise  | Imagenet-R  | CUB200  | CLRS25  | clrs25 |
> |:---:|:---:|:---:|:---:|---|
> | CE  | 70.6  | 79.4  | 67.4  | 76.3 |
> | aEPG  | 73.7  | 81.3  | 73.4  | 76.2 |
>
> **Scope (Class-incremental learning vs. Continual Detection/Segmentation)**. Our method is specifically tailored for class-incremental learning with a Markov Decision Process (MDP) formulation designed for classification tasks. In contrast, detection and segmentation require multi-step reward structures and pose fundamentally different challenges (e.g., background shift), which are typically studied as separate research problems. Our method is not directly relevant to those domains.
>
> **3. Orthogonality of Loss Investigation to Architecture Choice**
>
> Our work builds upon recent advances in continual PEFT, where CE loss is commonly adopted across various architectures such as LoRA, prompt tuning, and adapters.
>
> Our investigation is orthogonal to architectural design choices. It focuses solely on the loss function and its effect on continual learning. For instance, we use LoRA with a rank of 4 in our main experiments and additionally tested with a rank of 8. The performance trends remain consistent, indicating that our findings generalize across LoRA configurations. Table 2 in the paper also shows that the performance gains are applicable across other PEFT architectures (prefix and adapters).
>
>
> **4. Clarification on Reported Performance Improvements**
>
> In Table 2, the observed performance improvements of our method range from 1.5% to 3.7%, rather than the 0.5% mentioned in the reviewer’s comment. These gains are statistically significant (p < 0.001, paired t-test) and are also supported visually in Figure 2, which presents 95% confidence intervals.
>
> Once again, we thank the reviewer for their thoughtful feedback. We believe the clarifications and additional results we’ve provided strengthen the case for the relevance, robustness, and impact of our work.

---

> > ### Comment · Reviewer_L5mY · 2025-08-05
> > **Official Comment 1 by Reviewer L5mY**
> >
> > We thank the authors for their comprehensive rebuttal. I have carefully reviewed all initial reviews and the authors’ responses. Specifically, I acknowledge the authors’ sincere efforts in addressing the following points:
> >
> > - (In response to Reviewers Gcq7 and XVVp) The authors provided further clarification on why reinforcement learning (RL) is used for classification, and more specifically, why it is applied to continual learning.
> > - (In response to Reviewer Gcq7) The authors emphasized that approaches for continual learning with PEFT are situated in a second-stage fine-tuning setup, and not based on CE loss from scratch.
> > - (In response to Reviewers XVVp and cwyn) The authors demonstrated the robustness of the adaptive entropy annealing hyperparameters, including step size and the annealing coefficient.
> > - (In response to Reviewers XVVp and cwyn) The authors included empirical comparisons with recent RL-based methods, including PPO, and showed that their method performs favorably.
> > - (In response to Reviewer XVVp) The authors clarified that their method is designed and evaluated in the context of fully supervised learning with labeled data, and thus follows a different direction from semi- or unsupervised methods.
> >
> > ---
> >
> > Nevertheless, this reviewer, in line with Gcq7’s review, still has some reservations about whether the paper meets the acceptance threshold in terms of **overall quality**, **completeness of the literature review**, and **the sufficiency of empirical validation**.
> >
> > **A. On the Quality of Writing**
> >
> > This reviewer would like to revisit a point raised in an earlier review: “Some statements in the paper may *mislead readers unfamiliar* with the field into thinking that all past CL methods rely on replay.” To clarify, this comment was not intended to suggest that the authors need to include comparisons with additional continual learning methods beyond DualPrompt and L2P. Rather, the underlying concern is whether the paper, in its current form, can be read easily and naturally by researchers in the broader AI community. From this reviewer’s perspective, a good paper is one that can be clearly understood not only by AI researchers, but also, ideally, even by non-experts. It should offer broad knowledge and meaningful insights. However, this paper may still require significant improvements in writing clarity and contextual framing to meet that standard.
> >
> > **B. On the Sufficiency of Literature Review of Recent Work**
> >
> > This reviewer would respectfully like to clarify the intention behind the earlier comment: “Several statements like this raise concerns about whether the prior research has been sufficiently surveyed and effectively communicated.” This concern aligns with a similar point raised by Gcq7.
> >
> > The concern is about whether the most recent and relevant research has been appropriately contextualized within the rapidly evolving landscape of the field. The main baselines considered in this paper, DualPrompt (ECCV 2022) and L2P (CVPR 2022), are already somewhat dated. In a fast-moving field such as AI, carefully positioning the proposed work with respect to more recent developments is an important component of producing a high-impact paper.
> >
> > From this reviewer’s perspective, the introduction would have been more valuable if it had followed a clearer progression, such as:
> >
> > 1. What is continual learning?
> > 2. Among continual learning approaches, what strategies have been used for class-incremental learning?
> > 3. What is PEFT?
> > 4. Why is PEFT important in the context of continual learning?
> > 5. What are the key prior works that combine CL and PEFT?
> > 6. What are the limitations of those works?
> >
> > Such a step-by-step structure would allow the authors to clearly position their work and provide appropriate categorization and citation of recent publications in each sub-area. For instance, when compared with more recent papers (as listed in the table below), the number of reference works cited in this paper appears relatively small. This indirectly illustrates the concern about the depth of the literature review.
> >
> > Instead of providing such a structured overview of how recent studies have developed across these topics, the current paper’s narrative roughly follows this logic:
> >
> > 1. Traditional CL methods rely on memory replay.
> > 2. Some prior CL works have used PEFT.
> > 3. These works typically use CE loss.
> > 4. This paper proposes to use RL instead of CE loss.
> >
> > This progression leads to concern that substantial revisions may be required to make the narrative accessible and informative for the broader community.

---

> ### Comment · Reviewer_L5mY · 2025-08-05
> **Official Comment 2 by Reviewer L5mY**
>
> **C. On the Scalability of the Proposed Method**
>
> Reviewer cwyn previously noted that *“only ViT-B is used for experiments, and there is no evidence on CNNs or larger ViT models.”* Reviewer *L5mY* shares this concern. Despite access to high-end hardware (such as A6000 and A100 GPUs), the authors did not provide additional experiments involving larger architectures.
>
> From this reviewer’s perspective, while it may not be strictly necessary to evaluate the method on object detection tasks (which typically rely on CE loss), an extension to semantic segmentation would have been valuable. This is because semantic segmentation can be framed as pixel-level classification, which is closely aligned with the authors’ current setup. Demonstrating performance in this setting could significantly enhance the paper’s impact and generalizability.
>
> That said, as shown in the table below, this reviewer acknowledges that many works in this field have similarly limited benchmark and architecture coverage. However, in order to increase the influence of this work within the broader community, such scalability evaluations are viewed as an important next step.
>
> In fact, recent papers in self-supervised learning (such as, MoCo, SimCLR, SimSiam), image classification, and model architecture research (including CNNs, ViTs, Transformers, Mamba, and sliding window attention) often include comprehensive and extensive benchmark evaluations as a standard practice.
>
> |                                                              | Venue                 | #Reference Papers | Benchmarks                                            | Architectures      |
> | ------------------------------------------------------------ | --------------------- | ----------------- | ----------------------------------------------------- | ------------------ |
> | DualPrompt                                                   | ECCV 2022             | 69                | ImageNet-R, CIFAR100                                  |                    |
> | LAE                                                          | CVPR 2022             | 68                | ImageNet-R,  CIFAR100                                 |                    |
> | Continual Learning with Low Rank  Adaptation                 | NeurIPS Workshop 2023 | 38                | CORe50, DomainNet, Split CIFAR-100                    | ViT-B/16           |
> | Online-LoRA: Task-free Online Continual  Learning via Low Rank Adaptation | WACV 2025             | 103               | CORe50, CIFAR-100, ImageNet-R, ImageNet-S, CUB-200    | ViT-B/16, ViT-S/16 |
> | CL-LoRA: Continual Low-Rank Adaptation  for Rehearsal-Free Class-Incremental Learning | CVPR 2025             | 61                | CIFAR-100, ImageNet-R, ImageNet-A, VTAB               | ViT-B/16           |
> | SD-LoRA: Scalable Decoupled Low-Rank  Adaptation for Class Incremental Learning | ICLR 2025             | 44                | ImageNet-R, ImageNet-A, DomainNet, CIFAR-100, CUB-200 | ViT-B/16           |
> | This paper                                                   |                       | 32                | ImageNet-R, Food101, CUB200, CLRS                     | ViT-B/16           |
>
> ---
>
> Reviewer *L5mY* thanks the authors again for their thoughtful rebuttal. I would kindly encourage the authors to provide a concrete plan addressing points A, B, and C discussed above. It is understandable that the limited timeline may have made it difficult to conduct additional experiments related to point C, so this reviewer assures that the absence of such experiments will not be viewed negatively in the review score. Provided that a clear plan is presented, I remain open to further discussion with the reviewers and Area Chairs, and to revisiting my score.

---

> > ### Author Response · Authors · 2025-08-07
> >
> > Dear Reviewer,
> >
> > Thank you for your thoughtful and detailed comments, as well as your willingness to engage further with our work. We sincerely appreciate the time and care you’ve taken to provide such comprehensive feedback.
> >
> > We are encouraged that our rebuttal addressed many of your initial concerns. Below, we respond to the three major points you raised (A–C) and outline our plan to improve the paper accordingly.
> >
> >
> > **A. Writing Quality and Accessibility**
> >
> > We appreciate your emphasis on clarity and accessibility for a broader AI audience. We fully agree that a strong paper should not only communicate effectively with domain experts in continual learning, but also be readable and informative to researchers from other areas.
> >
> > To address this, we will revise the **Introduction** and **Related Work** sections. Specifically, we will expand the discussion to clearly introduce:
> >
> > * Continual learning and class-incremental learning;
> > * Parameter-efficient fine-tuning (PEFT);
> > * The importance of PEFT in continual learning;
> > * Existing methods that combine continual learning and PEFT;
> > * The limitations of these methods and our motivation for proposing an RL-based approach.
> >
> > Additionally, we will revise or remove statements that may inadvertently mislead readers, for example, those that overgeneralize the reliance on replay-based methods. We also plan to conduct a thorough review of the paper to improve clarity, coherence, and narrative flow.
> >
> > **B. On the Sufficiency of Literature Review of Recent Work**
> >
> > We would like to clarify that our submission already includes comparisons with more recent methods beyond DualPrompt and L2P, such as **LAE (ICCV 2023)** and **InfLoRA (CVPR 2024)**. That said, we acknowledge that some newer works were published after the submission deadline.
> >
> > In the revised version, we will expand the **Related Work** section to include discussions of these and other recent contributions, such as CL-LoRA (CVPR 2025). We will also update our narrative to more effectively position our method within the evolving landscape of continual learning and PEFT.
> >
> > **C. On the Scalability of the Proposed Method**
> >
> > Our current experiments follow standard practice in the field, using ViT-B/16 and image classification benchmarks for comparison. However, we agree that additional scalability evaluations would strengthen the paper.
> >
> > To this end, we have conducted new experiments using **ViT-Large (ViT-L/16)** with two LoRA placement strategies (in the first 5 and first 10 transformer blocks). We observed consistent performance trends, with **aEPG significantly outperforming CE**, further validating the robustness of our method on larger models.
> >
> > **Rebuttal Table 1. Additional Experiment Results with ViT-L/16**
> > |   | Imagenet-R 200  | CUB200 | CLRS25   |
> > |:---:|:---:|:---:|:---:|
> > | LoRA Block: 0-4 |  |  |  |
> > | CE | 76.4 | 81.0 | 73.1 |
> > | aEPG | 79.3 | 82.8 | 74.6 |
> > | LoRA Block: 0-10 |  |  |  |
> > | CE | 78.0 | 82.6 | 73.5 |
> > | aEPG  | 80.2 | 84.6 | 76.8 |
> >
> > As for **continual segmentation**, it presents unique challenges such as background shift. While continual semantic segmentation is typically addressed as a separate subfield, we agree that extending our framework to this domain is a meaningful next step. We are currently exploring how to formulate a suitable MDP (i.e., the design of state, action and reward) for pixel-level classification and plan to pursue this in future work.
> >
> > Thank you again for your constructive feedback.

---

> ### Comment · Reviewer_L5mY · 2025-08-07
>
> Dear authors,
>
> Thank you for providing a concrete revision plan and for including additional experimental results with ViT-L. I appreciate the author's the thoughtful direction in which you are planning to improve the paper. **I am pleased to raise my score by one point in recognition of their efforts and the potential of thier method**. That said, I still believe the initial manuscript requires notable restructuring and clarity improvements. This remains an important factor in my overall assessment. At present, I am continuing discussions with other reviewers and am sincerely taking all their feedback into account with an open mind. I commit to actively participating in the discussion and carefully considering all perspectives before finalizing my score.
>
> Best regards,
> Reviewer L5mY

---

### Official Review · Reviewer_cwyn · 2025-07-03

**Clarity:** 3
**Significance:** 3
**Originality:** 3
**Rating:** 5
**Confidence:** 4

**Summary:**

The submission reformulates image classification as a one-step MDP, treats the 0-1 accuracy as a reward, and derives a low-variance Expected Policy Gradient (EPG) estimator. The authors argue that, compared with cross-entropy (CE), EPG implicitly down-weights uncertain samples and drives the soft-max to lower entropy, and thus leads to better continual fine-tuning. They further propose an adaptive EPG (aEPG) to anneal from CE to EPG during training. Experiments on four classification benchmarks with Vision-Transformer backbones and several PEFT modules show aEPG outperforming CE and a few entropy-increasing losses. This work encourages the community to reconsider loss-function choices for fine-tuning. However, I am concerned about the novelty and the empirical scope of the current manuscript. I thus recommend borderline rejection, but I am happy to increase the score if concerns are addressed.

**Questions:**

1.	Figure 1 demonstrates that EPG converges much faster than CE. However, their initial entropy loss on the next data split is close. Does EPG also demonstrate better generalization along with faster convergence, or does the faster convergence mainly stem from overfitting? It is suggested to plot the accuracy with the entropy in Figure 1.
2.	CE loss is known to exhibit overconfidence. Since EPG further lowers the entropy, would EPG exacerbate the overconfidence issue? It is suggested to compare EPG and CE on the expected confidence error  (ECE) metric.

**Ethical Concerns:**

["NO or VERY MINOR ethics concerns only"]

**Final Justification:**

Thank you for the authors' outstanding rebuttal. It really addresses most of my concerns. I believe that including these discussions in the revision could immensely improve the manuscripts, and I will increase my score to acceptance.

**Limitations:**

Please refer to the weaknesses.

**Quality:**

3

**Strengths And Weaknesses:**

**Strengths**:

1.	The authors provide clear motivation to use reinforcement learning with the 0-1 accuracy as rewards, and the paper is well organized. Moreover, the MDP formulation, gradient derivation, and entropy discussion are easy to follow.
2.	The study shows that lower predictive entropy benefits continual fine-tuning of pretrained ViTs, challenging the common “high-entropy helps” view from scratch training.
3.	Experiments on four classification benchmarks with Vision-Transformer backbones and several PEFT modules show aEPG outperforming CE and a few entropy-increasing losses.

**Weaknesses**:

1.	Unconvincing novelty. The idea of reward-based fine-tuning for vision models was already introduced in [1], which uses reinforcement learning to optimise task-specific metrics. The current manuscript lacks a thorough comparison and discussion with [1] to clearly demonstrate their superiority and novelty.
2.	Insufficient evaluations. i) Task coverage is narrow, only class-incremental image classification is considered, omitting other vision tasks (e.g., detection or segmentation) and continual-learning settings; ii) Metric coverage is equally limited, while claiming to mitigate forgetting, the study omits standard CL metrics such as forward/backward transfer and forgetting rate; iii) From the model perspective, only ViT-B is used for experiments, there is no evidence on CNNs and larger ViTs.
3.	Scalability concerns. As From Table 1, the advantage of aEPG over CE loss quickly diminishes with more learning steps, i.e, from +2.2% at $A_5$ to +0.9% at $A_{20}$ with LoRA, raising doubts about scalability to longer streams.
4.	Unclear benefits. Although the study is motivated as "using RL", the core benefit reduces to explicit entropy control. A much simpler CE + Entropy-Penalty baseline performs competitively, and this EP loss often surpasses EPG on the same datasets as in Table 2, weakening the value of the proposed reinforcement formulation.
5.	Potential fragility. By down-weighting uncertain predictions, EPG favours exploitation over exploration, risking over-fitting and sensitivity to label noise. The mild class imbalance in ImageNet-R is insufficient to test this, and stronger noise and imbalance scenarios are needed to validate robustness across fine-tuning settings.

[1] Tuning Computer Vision Models With Task Rewards. ICML'23.

---

> ### Author Rebuttal · Authors · 2025-07-31
>
> We thank the reviewer for taking the time to read and critically evaluate our work. We are encouraged by the reviewer’s appreciation of our _clear motivation_ and recognition that _the paper challenges the commonly held belief that “high-entropy helps.”_ We also value the acknowledgement that our work invites the community to reconsider loss function choices in the context of continual fine-tuning. Below, we respond to the reviewer’s concerns regarding novelty and experimental scope:
>
> **1. Novelty and Comparison with [1]**
>
> We appreciate the reviewer pointing us to [1] and commit to including a dedicated paragraph in the revised version to explicitly discuss its relationship to our work and add the experiment comparison result.
>
> While [1] applies sampling-based policy gradient methods to detection and segmentation tasks, our work focuses on classification and proposes a tailored one-step MDP formulation with a fully observable reward model. This formulation allows us to derive a low-variance policy gradient estimator, eliminating the need for sampling actions and resulting in a more stable and efficient optimization process.
>
> To highlight this distinction, we include an empirical comparison between REINFORCE (following Algorithm 2 in [1]) and our Expected Policy Gradient (EPG). As expected, EPG consistently achieves better performance across different benchmarks.
>
> **Additional Result Table 1: Comparison with sampling-based policy gradient**
> |  | cub200 | clrs25 | imagenet-r |
> |---|---|---|---|
> | EPG | 84.5 | 74.6 | 75.1 |
> | REINFORCE | 75.3 | 73.4 | 73.7 |
>
> Conceptually, our proposed MDP formulation and EPG unify reinforcement learning (RL) and maximum likelihood estimation (MLE) within a single optimization framework, revealing a deeper theoretical connection between the two, as shown in Eq 10. In contrast, [1] uses separate optimization strategies for RL and MLE (as seen in their Algorithms 1 and 2).
>
> Other differences between our work and [1] include different problem settings (continual learning vs. fine-tuning) and different tasks (classification vs. detection/segmentation).
>
> **2. Robustness to overfitting risk and label noise**
>
> Contrary to the reviewer's concern about increased overfitting, our findings indicate that EPG/aEPG is actually less prone to overfitting than cross-entropy. This is because EPG prioritizes easier samples during gradient updates, promoting the learning of general patterns in the data. In contrast, CE emphasizes hard or uncertain samples, which makes it more susceptible to overfitting to samples that contain noise.
>
> To support this claim, we conducted additional experiments under label noise conditions (20% symmetric label noise). Our findings show that aEPG significantly outperforms CE in these settings, indicating greater robustness.
>
> **Additional Result Table 2: Performance under noisy label setting**
> | 20% Noise  | Imagenet-R  | CUB200  | CLRS25  | clrs25 |
> |:---:|:---:|:---:|:---:|---|
> | CE  | 70.6  | 79.4  | 67.4  | 76.3 |
> | aEPG  | 73.7  | 81.3  | 73.4  | 76.2 |
>
>  All reported results are averaged over three runs, and the results are statistically significant (paired t-test, p < 0.0001).
>
> Theoretically, reinforcement learning is inherently designed for stochastic reward signals, making it naturally tolerant to noise. It can also be rigorously proved that the proposed EPG objective is noise-tolerant under symmetric or uniform noise, if the noise rate $n<1-1/K$, where K is the class number, i.e., that the global minimizer of EPG under label noise remains the same as the minimizer under clean labels.
>
>  **3. Calibration performance**
>
> We agree that since aEPG produces lower output entropy compared to CE, aEPG can make the model more overconfident. However, this does not necessarily imply poorer calibration. As shown in "Do Not Be Afraid of Overconfidence"[1], overconfidence may not hurt final calibration performance if post-hoc calibration is allowed. And in some cases more overconfident training paradigm (e.g., "inverse focal loss") can even lead to more calibratable models after the post-doc calibration stage.
>
> Following the insight in [1], we applied post-doc calibration (temperature scaling on a separate validation set) to both CE and aEPG. After calibration, aEPG achieves comparable or even better Expected Calibration Error (ECE) results than CE.
>
> **Additional Result Table 3: Calibration performance**
>  | ECE performance  | Imagenet-R  | CUB200  | CLRS25  |
> |:---:|:---:|:---:|:---:|
> | CE  | 0.084  | 0.037  | 0.049  |
> | aEPG  | 0.078  | 0.034  | 0.033  |
> | P value  | <0.001  | 0.118 (insignificant)  | <0.001  |
>
> [1] "Rethinking Calibration of Deep Neural Networks: Do Not Be Afraid of Overconfidence" NeurIPS 2021
>
> **4. Performance advantage persists for longer streams**
>
> To address the reviewer's concern about the performance over long CL tasks, we conducted additional experiments by splitting the original benchmarks into longer task sequences. The results confirm that aEPG's performance advantage significantly persists in extended continual learning settings (paired t test: p<0.001).
>
> **Additional Result Table 4: Evaluation with longer CL tasks**
> |   | Imagenet-R 200  | CUB200   | CLRS25   |
> |:---:|:---:|:---:|:---:|
> |   | 10 to 40 tasks  | 10 to 40 tasks  | 5 to 12 tasks  |
> | CE  | 69.7  | 74.0  | 55.6  |
> | aEPG  | 70.3  | 75.1  | 56.6  |
>
> **5. Entropy control: aEPG vs. Entropy Penalty**
>
> We appreciate the reviewer’s attention to the role of entropy. Indeed, a key message of this work is to highlight the impact of entropy reduction in continual fine-tuning. To our knowledge, this aspect has not been systematically explored before in continual fine-tuning. Our findings challenge the prevailing belief that high-entropy predictions are beneficial: Table 2 demonstrates that increasing entropy tends to hurt performance, while reducing entropy leads to better CL outcomes, opening a promising direction for future research on entropy-control strategies.
>
> In terms of entropy control mechanisms, rather than applying explicit entropy penalties, aEPG provides a novel mechanism to reduce entropy based on sample weighting. We demonstrate that emphasis on harder examples (via importance sampling) in the gradient leads to effective entropy reduction. Empirically, aEPG outperforms standard entropy-penalty baselines in three out of four benchmarks, except for CUB-200. We hypothesize that this may be due to the low intra-class variance common in fine-grained classification tasks, where the entropy reduction achieved by importance sampling is less pronounced.
>
> Once again, we thank the reviewer for their time and effort, which will help us strengthen the paper.

---

> ### Comment · Reviewer_cwyn · 2025-08-02
>
> Thank you for your outstanding rebuttal :). It really addresses most of my concerns. I believe that including these discussions in the revision could immensely improve the manuscripts, and I will increase my final score accordingly.
>
> For follow-up, I still believe that "CE+entropy penalties" (termed EP)  and the proposed EPG conceptually are very close. EPG can be boiled down to "reverse-KL + entropy penalties", where "CE+entropy penalties" is equivalent to "forward-KL + entropy penalties". In Table 2,  EP outperforms EPG across all benchmarks and is likely to further benefit from adaptive entropy annealing due to its similarity with EPG, which may outperform aEPG.
>
> Personally, the paper is clearly motivated and technically sound, but positioning EPG/aEPG as a superior alternative may be a little overstated, since the main empirical lesson is the explicit entropy control. I would advice reframing the takeaway accordingly (e.g., “entropy control matters for continual fine-tuning; aEPG is one effective way to achieve it”) and tempering claims about RL-specific unless with more results that demonstrate the distinct benefit of aEPG over EP(+annealing).

---

> > ### Author Response · Authors · 2025-08-04
> >
> > Thank you very much for your thoughtful and encouraging feedback! We’re glad our rebuttal helped clarify most of your concerns. We truly appreciate your positive assessment and the time and effort you’ve invested in providing valuable suggestions that help improve this work. We will make sure to incorporate these discussions in the revision.
> >
> > We agree that EP (i.e., CE + entropy penalties) and EPG are closely related, and we appreciate your insightful observation regarding the duality: EP as forward-KL + entropy penalties, and EPG as reverse-KL + entropy penalties.
> >
> > Regarding performance, you're absolutely right that EP shows strong results in Table 2, and we acknowledge that it may benefit further from adaptive entropy scheduling. We initially explored this direction but found limited improvements. During the rebuttal phase, we re-ran experiments on CLRS, ImageNet-R, and Food101 using a similar annealing schedule for aEPG and aEP. The results confirm that aEPG consistently outperforms adaptive EP (aEP). We suspect that aEPG’s performance advantage in these three datasets may stem from its alignment with the 0-1 classification accuracy objective during the later stages of training.
> >
> > That said, we agree that the relative strengths of EP and aEPG likely depend on dataset characteristics and the specific entropy scheduling strategy. This is an important point that deserves a more nuanced discussion, and we will make sure to highlight it in the revised manuscript.
> >
> > Your suggestion to reframe the main takeaway is well taken. We will revise the narrative to emphasize the broader insight that explicit entropy control is important for continual fine-tuning and position aEPG as one effective approach among others.
> >
> > Once again, thank you for your constructive feedback. It has been instrumental in helping us refine both the framing and clarity of our work.
> >
> > |   | Imagenet-R 200  | Food101 | CLRS25   |
> > |:---:|:---:|:---:|:---:|
> > | CE | 74.3 | 82.9 | 72.6 |
> > | EP | 75.0 | 83.7 | 76.3 |
> > | aEP | 74.7 | 83.7 | 76.3 |
> > | aEPG  | 75.8 | 84.5 | 76.8 |

---

> > > ### Comment · Reviewer_cwyn · 2025-08-04
> > >
> > > Thank you for your new responses! I believe the empirical insight on explicit entropy control is immensively important for continual finetuning, and may further connect various research directions, e.g., reinforcement learning, test-time entropy minimization, etc.
> > >
> > > In light of the impressive rebuttals provided and the potential impact this work could have on the community, I raise my initial score to Accept and wish the authors the best with their paper.

---

> > > > ### Author Response · Authors · 2025-08-07
> > > >
> > > > Dear reviewer,
> > > >
> > > > Thank you for your encouraging feedback and for engaging deeply with our work. We greatly appreciate your recognition of the broader significance of ​​explicit entropy control​​ in continual fine-tuning.
> > > >
> > > > We’re also sincerely grateful for your updated evaluation and your support for the paper. Your thoughtful comments have been invaluable in refining our work.
> > > >
> > > > Thank you once again for your time, expertise, and kind words. Your engagement has made this work stronger, and we truly appreciate it.

---

### Official Review · Reviewer_XVVp · 2025-07-03

**Clarity:** 2
**Significance:** 3
**Originality:** 3
**Rating:** 4
**Confidence:** 3

**Summary:**

The paper reformulates the classification problem through reinforcement learning to solve the continual fine-tuning task. The authors frame classification as a one-step Markov Decision Process (MDP) and introduce Expected Policy Gradient (EPG) to directly minimize the 0-1 loss. They also propose an adaptive entropy annealing strategy (aEPG) to combine the strengths of both EPG and cross-entropy for continual fine-tuning. Experiments on several continual learning benchmarks demonstrate the effectiveness of the proposed method.

**Questions:**

1. It is interesting that upweighting confident predictions and downweighting uncertain ones improves the final results for continual fine-tuning. Are there any analyses on the reasons?

2. I'm also curious whether the proposed method can potentially or theoretically be combined with semi-supervised or unsupervised settings.

3. How does the proposed method solve or reduce the catastrophic forgetting problem in continual learning?

**Ethical Concerns:**

["NO or VERY MINOR ethics concerns only"]

**Final Justification:**

The authors' rebuttal solved most of my concerns and I do think the idea of the paper is interesting. By considering the novelty of the paper as well as some problems raised by the other reviewers, I will keep my initial score but I believe the paper is above the acceptance bar.

**Limitations:**

yes

**Quality:**

3

**Strengths And Weaknesses:**

Strengths:

1. The idea that formulates 0-1 classification as a reinforcement learning problem is interesting.

2. Experiments demonstrate the effectiveness of the proposed method.

Weaknesses:

1. The motivation is not such clear. Why introduce reinforcement learning to assist the CE-based classification task? What problem of CE-based continual learning does the proposed RL method solve?

2. Is the adaptive entropy annealing sensitive to the hyperparameter $\tau$? In the paper, it is just set to 6 without any empirical or theoretical analyses.

3. Most of the methods in the comparison tables are common classification methods. Lack of comparisons with recent RL-based methods.

4. The method is proposed only for the fully supervised setting. Applications in semi-supervised or unsupervised settings are more realistic but have not been explored in the paper, which limits its scalability.

---

> ### Author Rebuttal · Authors · 2025-07-31
>
> We thank the reviewer for taking the time to read and critically assess our paper.
>
> We are encouraged that the reviewer found the core idea of formulating 0-1 classification as a reinforcement learning problem _interesting_. We address the questions point by point as follows.
>
> **1.The motivation of EPG/aEPG for continual learning**
>
> Thank you for raising the question regarding the motivation behind EPG. In retrospect, we agree that this aspect could have been communicated more clearly, and we will revise the paper accordingly to make this more explicit.
>
> Our central motivation is that reinforcement learning enables direct optimization of the true classification objective: maximizing accuracy (i.e., minimizing the 0-1 loss). While the RL method (i.e. EPG) can directly minimize 0-1 loss, our analysis reveals that cross-entropy (CE) loss introduces an _exploration bias_, which can exacerbate forgetting.
>
> aEPG mitigates this issue by leveraging CE’s exploratory behavior early in training, and gradually annealing toward 0–1 loss optimization as learning progresses. This enables aEPG to maintain CE’s adaptability (plasticity) while gaining the stability benefits of directly optimizing for accuracy, resulting in a more favorable stability-plasticity trade-off.
>
> Concretely, in parameter space, CE’s exploration bias is evident in its gradient expression:
> $ g_{ce}=-1/p_\theta(y|x)g_{epg}$. This means CE focuses on uncertain or hard examples, which yield high information gain (or "surprisal"), as the model must correct large prediction errors. While this drives adaptation to a new task, it can also disrupt existing knowledge, increasing forgetting. In contrast, EPG/aEPG focuses on easier samples that already align well with the model’s predictions, resulting in lower surprisal and less disruptive updates. In action space, CE’s exploration bias manifests as higher output entropy than 0-1 loss optimization.
>
> We hope this clarifies the motivation behind aEPG and its observed benefits. We are happy to provide further elaboration if helpful.
>
> **2. Potential Applications in Semi-/Unsupervised Settings**
>
> Our current MDP formulation is specifically designed for supervised learning, where class labels are used to define the reward. Extending this framework to semi-supervised or unsupervised learning would require a new MDP formulation—a direction we agree is both interesting and promising for future work.
>
> One possible extension would be to design RL tasks based on self-supervised objectives. For example, SimCLR can be reframed as a classification task: given an anchor sample and a batch of candidates, the model must select the correct augmented version of the anchor. A reward function can then assign 1 for correct selection and 0 otherwise, enabling a policy gradient formulation for contrastive learning.
>
> **3. Choice of hyperparameter $\tau$**
>
> The hyperparameter $\tau$ controls the annealing schedule for transitioning from CE loss ($\alpha=1$) to EPG loss ($\alpha=0$). We use a sigmoid-based schedule and $\tau$ determines the range of the transition: when $\tau=6$, we have $\alpha_{start}=sigmoid(6) =0.998 $ and $\alpha_{end}=sigmoid(−6) = 0.002$. This provides a smooth and effective transition (approximately from 1 to 0) over the training period.
>
> In response to the reviewer's question, we tested $\tau$ values of 4, 6, and 8 and found that they achieve similar performances.
>
> **Additional Result Table 1: The effect of $\tau$**
> |  | $\alpha_{start}$ |  $\alpha_{end}$| imagenet-r | clrs25 |
> |---|---|---|---|---|
> | $\tau=4$ | 0.9820 | 0.0180 | 75.9 | 76.3 |
> | $\tau=6$ | 0.9975 | 0.0025 | 75.9 | 76.2 |
> | $\tau=8$ | 0.9997 | 0.0003 | 75.6 | 76.5 |
>
> Please note that the paper also contains an ablation study on different annealing strategies (sigmoid, cosine, linear) and found that performance remained robust across these schedules (see Section 4.2 and Figure 3).
>
> **4. Comparison with recent RL methods**
>
> Recent RL-based approaches to classification typically rely on sampling-based policy gradient methods (e.g., REINFORCE, PPO), which are more suited to multi-step MDPs with unknown reward structures. Our formulation, by contrast, is a one-step MDP with a fully known reward model.
>
> This allows EPG to compute a closed-form policy gradient, avoiding the high variance typically associated with sampling-based approaches. As a result, we expect—and observe—better performance from EPG compared to methods like REINFORCE.
> To support this point, we added comparisons with REINFORCE. As shown below, EPG significantly outperforms REINFORCE
>  (p<0.001).
>
> **Additional Result Table 2: Comparison with sampling-based policy gradient$**
> |  | cub200 | clrs25 | imagenet-r |
> |---|---|---|---|
> | EPG | 84.5 | 74.6 | 75.1 |
> | REINFORCE | 75.3 | 73.4 | 73.7 |
>
> Once again, we thank the reviewer for their time and effort, which will help us improve the clarity and impact of the paper.

---

> > ### Comment · Reviewer_XVVp · 2025-08-05
> >
> > I thank the authors for their rebuttal. It solved most of my concerns. I really like the possible extensions of designing RL tasks based on self-supervised objectives.
> >
> > In addition, I am still curious on how the proposed method solve or reduce the catastrophic forgetting problem in continual learning, intuitively or theoretically. I believe adding some discussions about this problem will also improve the paper.

---

> > > ### Author Response · Authors · 2025-08-07
> > >
> > > Dear Reviewer,
> > >
> > > Thank you for your thoughtful and encouraging feedback. We appreciate your constructive comments and are pleased that our rebuttal solved most of your concerns.
> > >
> > > Regarding your question about catastrophic forgetting in continual learning, our method mitigates forgetting by explicitly regularizing information gain during gradient updates. Intuitively, when learning new information, a large information gain (or surprisal) suggests that the model is encountering information significantly different from its existing knowledge. Therefore, such learning updates can be disruptive, potentially overwriting previously learned representations.
> > >
> > > To address this, the proposed gradient prioritizes easier samples, those on which the model already performs well. These samples naturally induce smaller information gains per update, which helps reduce disruptive parameter changes and preserves prior knowledge.
> > >
> > > We fully agree that a more detailed discussion of this mechanism would strengthen the paper. In the revised version, we will expand our analysis in Section 3.3 to better clarify the connection between information gain regularization and forgetting mitigation.
> > >
> > > More broadly, our empirical findings in Table 2 also suggest a link between entropy minimization and forgetting. We believe that exploring the theoretical relationship between information gain, entropy minimization, and catastrophic forgetting represents a promising direction for future research.
> > >
> > > Once again, we sincerely appreciate your time and valuable suggestions. Your insights have helped improve this work, and we look forward to addressing your points in the final manuscript.

---

> > > > ### Comment · Reviewer_XVVp · 2025-08-07
> > > >
> > > > I thank the authors for the further answers. I do not have any other questions. I will keep my initial score (4, Borderline Accpet). I wish the authors all the best during the rebuttal. Thank you.

---

### Official Review · Reviewer_Gcq7 · 2025-07-06

**Clarity:** 3
**Significance:** 2
**Originality:** 2
**Rating:** 4
**Confidence:** 4

**Summary:**

This paper studies a RL-based, policy-gradient approach to directly optimize for the 0-1 loss in continual classification problems, as opposed to standard continual cross entropy loss minimization; showing differences in exploitation and exploration between both training approaches. By introducing a custom annealing heuristic effectively transitioning between both approaches, the authors are able to achieve performance improvements on four smaller, standard continual learning benchmarks.

**Questions:**

See Weaknesses. In essence, it would be crucial if the authors could clearly highlight what motivated the use of EPG / aEPG for the actual task of continual learning, better discuss existing works in the continual learning domain, and extend their experiments over more benchmarks to more clearly see experimental significance.

**Ethical Concerns:**

["NO or VERY MINOR ethics concerns only"]

**Final Justification:**

The authors have clarified the large majority of my raised issues and concerns, alongside additional LAE results. Consequently, I've raised my score to suggest acceptance.

**Limitations:**

Explicitly addressed in Discussions.

**Quality:**

2

**Strengths And Weaknesses:**

__Strengths.__

* The proposed EPG method is very sensible, and the conducted experiments into exploitation versus exploration make sense.

__Weaknesses.__

Unfortunately, I have some larger issues regarding the motivation and relevance of the proposed approach to the continual learning domain:
* It does not make much sense to me as to why an RL-based approach, even if it optimized the 0-1 objective, would counter catastrophic forgetting in CL. The insights provided comparing CE and EPG are interesting, but essentially centered about general difference between both approaches, with no particular application to the continual learning domain as far as I can understand it. It would be great if the authors could provide some context here.
* The lack of relation to continual learning is also reflected in the very limited discussion of continual learning works, effectively revolving around a single survey, and discounting other existing regularization-based techniques; and the studied continual learning scenario is highly limited to vision-based classification models on small-scale benchmarks fewer in numbers than commonly studied in comparable works such as e.g. in LAE, the utilized base method. Already when studied on four benchmarks and contrasted against simple continual CE (which could easily be augmented with commonly deployed momentum-based interpolation regularization techniques), gains are also not entirely convincing. It would be great if the authors could better highlight the experimental relevance.

Fundamentally, the paper is evaluated as a continual learning paper; but the method and even experiments are insufficient related / provide insufficient coverage to be really accounted as such at this point.

---

> ### Author Rebuttal · Authors · 2025-07-31
>
> We thank the reviewer for taking the time to read and critically assess our paper.
>
> We are encouraged that the reviewer found the proposed EPG method _very sensible_. We address the questions regarding the motivation and experiments as follows.
>
> **1. The motivation of EPG/aEPG for continual learning**
>
> Thanks for your question about the motivation of EPG. On reflection, we could give more context about the motivation behind EPG. We can commit to adding more discussion on this.
>
> Recent works in continual learning with pretrained models (e.g., L2P, DualPrompt, LAE, etc) all use local cross entropy (CE) as the training loss. Our analysis reveals that local CE exhibits an inherent _exploration bias_, which can exacerbate forgetting.
>
> Concretely, in the parameter space, CE’s exploration bias manifests as an emphasis on uncertain/hard samples in its gradient, i.e., $g_{ce}=-1/p_\theta(y|x)g_{epg}.$ Learning hard/uncertain samples leads to high information gain (or "surprisal"), as the model must correct large prediction errors. While this drives adaptation to a new task, it can also disrupt existing knowledge in order to accommodate the high information gain. In contrast, EPG/aEPG focuses on easier samples that already align well with the model’s predictions, resulting in lower surprisal and less disruptive updates. In action space, CE’s exploration bias manifests as higher output entropy than 0-1 loss optimization.
>
>
> aEPG addresses this issue by retaining CE’s exploratory behavior during the early phase of learning and gradually reducing exploration bias in the later training stage. This allows the model to benefit from CE’s plasticity while transitioning toward the stability of 0-1 loss optimization, thereby improving the overall stability-plasticity trade-off.
>
> We hope this clarifies the motivation behind aEPG and its observed benefits. We are happy to provide further elaboration if helpful.
>
> **2. Comparison with Related Continual Learning Works**
>
>  Replay-based (e.g., ER, DER++) and regularization-based (e.g., LwF, EWC) methods were originally designed for continual learning in training-from-scratch scenarios, typically using global CE loss. These methods are less prevalent in the context of continual parameter-efficient fine-tuning (PEFT) with pretrained models. Empirical studies in early continual finetuning works (e.g., DualPrompt [1], L2P [2]) have established that these methods (e.g., LwF, EWC) often underperform compared to continual PEFT-specific approaches like L2P and DualPrompt.
>
> Therefore, our comparisons focus on state-of-the-art **continual PEFT techniques**, including DualPrompt, LAE, LoRA, and InferLoRA.
>
> [1] "Dualprompt: Complementary prompting for rehearsal-free continual learning", ECCV 2022.
>
> [2] "Learning to prompt for continual learning" (L2P) CVPR 2022
>
> **3. Orthogonality of Loss Investigation​​**
>
> Our study (Table 2 in the paper) examines loss design independently of the architecture design of different continual PEFT techniques. Since methods like DualPrompt (with prompt-tuning module) and LAE (with old and new PEFT modules) all use CE loss, replacing their loss with aEPG is straightforward. Table 1 in the paper demonstrates that LAE (momentum-based update) combined with aEPG achieves significant gains for ImageNet-R. This finding is consistent with other benchmarks (see below).
>
> **Additional Result Table 1: Comparison with LAE on the other three benchmarks**
> |   | CUB200  | FOOD101  | CLRS25  |
> |:---:|:---:|:---:|:---:|
> | LAE+CE  | 82.7  | 83.3  | 72.9  |
> | LAE+aEPG  | 84.1  | 84.8  | 74.8  |
>
>  All reported results are averaged over three runs, and the improvements are **statistically significant (paired t-test, p < 0.0001)**.
>
>
> **4. Clarifications​​ on the benchmarks**
>
> We carefully reviewed our benchmark setup and compared it to related works such as LAE. Contrary to the reviewer’s concern, our work does not use smaller benchmarks. In fact, we evaluate on a broader and more challenging set.
>
> |   | Benchmarks  |
> |:---|:---|
> | DualPrompt  | ImageNet-R (30k samples), CIFAR100  |
> | LAE  | ImageNet-R (30k samples), CIFAR100  |
> | Ours work  | ImageNet-R (30k samples), Food101 (101k samples), CUB200, CLRS  |
>
> While LAE and DualPrompt evaluate on two datasets (CIFAR-100 and ImageNet-R), our study includes four more demanding benchmarks: ImageNet-R, Food101, CUB-200, and CLRS. Food101 introduces a significantly larger dataset size, while CLRS involves remote sensing data, further increasing diversity and difficulty.
>
> If we have misunderstood the reviewer’s concern about the benchmarks, we would appreciate further clarification.
>
> Once again, we thank the reviewer for their time and effort, which will help us improve the paper.

---

> > ### Comment · Reviewer_Gcq7 · 2025-08-05
> > **Response to Rebuttal**
> >
> > I thank the authors for their detailed rebuttal, which as clarified the large majority of my raised issues and concerns. As a result, I'm happy to raise my score and recommend acceptance, particularly with the inclusion of new LAE results.

---

> > > ### Author Response · Authors · 2025-08-07
> > >
> > > Dear reviewer,
> > >
> > > Thank you for your thoughtful and encouraging feedback. We’re glad that the rebuttal has addressed most of your concerns. We greatly appreciate your constructive feedback, which has been instrumental in improving our submission. In the revised version of the paper, we will include new results for LAE and extend it with additional PEFT modules (e.g., adapter and prefix).
> > >
> > > We’re also sincerely grateful for your updated evaluation and your support for the paper.

---

### Note · Authors · 2025-08-13

We thank the reviewers for their constructive feedback and active engagement throughout the process. The discussion phase has been particularly productive, with three reviewers (Gcq7, cwyn, XVVp) explicitly recommending acceptance following our rebuttal, while Reviewer L5mY raised their score in recognition of our clarifications and additional results.

Our rebuttal addressed the key points raised:

- **Motivation & CL relevance**. Through gradient and entropy analysis, we reveal that CE’s exploration bias can exacerbate forgetting. The proposed aEPG mitigates this by gradually annealing from CE to direct 0–1 loss optimization, improving the stability–plasticity trade-off in continual fine-tuning.

- **Extended comparisons**. We extended evaluations to other RL-based methods (e.g., REINFORCE) and strong baselines such as LAE and CE+(adaptive) Entropy Penalty, demonstrating aEPG’s consistent advantage across multiple datasets.

- **Robustness & scalability**.  ​​Additional experiments confirmed aEPG’s robustness to label noise and scalability to larger ViT-L/16 models, where it maintains clear improvements over CE.

- **Novel insight**. Reviewers particularly emphasized our broader takeaway: the critical importance of explicit entropy control in continual fine-tuning (as noted by Reviewer cwyn). aEPG provides a theoretically grounded RL mechanism to achieve this control effectively.

- **Clarifications & writing**. We committed to expanding the introduction and related work for clearer framing, and improved accessibility for a wider AI audience.

**Impact and Contributions**. Empirically, aEPG delivers statistically significant improvements for state-of-the-art continual fine-tuning methods (e.g., LAE) across four challenging benchmarks (ImageNet-R, Food101, CUB-200, CLRS). Theoretical connections between RL and MLE, along with noise-tolerance properties, further bridge multiple research directions, including reinforcement learning, continual learning, and entropy minimization.

We believe the final manuscript will make a meaningful contribution to the field by
- proposing **a ​​unified loss framework​​ that bridges RL and CE optimization** through a novel classification MDP formulation and expected policy gradient,
- **highlighting the underexplored role of entropy control** in continual fine-tuning.

We appreciate the AC’s time and consideration of this work.

---

### Decision · Program_Chairs · 2025-09-17

**Decision:**

Reject

**Comment:**

The paper had mixed reviews, with one reviewer accepting it, two weekly rejecting it and one rejecting it. There was a long discussion with some of the reviewers that led the authors to add many new results during the rebuttal and improve the paper. Nevertheless, the reviewers still find it borderline and the paper does not meet the NeurIPS bar. I suggest the authors to reflect the feedback from the rebuttal, polish the paper, and submit it to a new venue.